# Budgeted Reinforcement Learning in Continuous State Space

**Nicolas Carrara**[*]
SequeL team, INRIA Lille – Nord Europe[†]
`nicolas.carrara@inria.fr`

**Edouard Leurent**[*]
SequeL team, INRIA Lille – Nord Europe[†]
Renault Group, France
`edouard.leurent@inria.fr`

**Romain Laroche**
Microsoft Research, Montreal, Canada
`romain.laroche@microsoft.com`

**Tanguy Urvoy**
Orange Labs, Lannion, France
`tanguy.urvoy@orange.com`

**Odalric-Ambrym Maillard**
SequeL team, INRIA Lille – Nord Europe
`odalric.maillard@inria.fr`

**Olivier Pietquin**
Google Research - Brain Team
SequeL team, INRIA Lille – Nord Europe[†]
`pietquin@google.com`

## Abstract

A Budgeted Markov Decision Process (BMDP) is an extension of a Markov Decision Process to critical applications requiring safety constraints. It relies on a notion of risk implemented in the shape of a cost signal constrained to lie below an – adjustable – threshold. So far, BMDPs could only be solved in the case of finite state spaces with known dynamics. This work extends the state-of-the-art to continuous spaces environments and unknown dynamics. We show that the solution to a BMDP is a fixed point of a novel Budgeted Bellman Optimality operator. This observation allows us to introduce natural extensions of Deep Reinforcement Learning algorithms to address large-scale BMDPs. We validate our approach on two simulated applications: spoken dialogue and autonomous driving[3].

## 1 Introduction

Reinforcement Learning (RL) is a general framework for decision-making under uncertainty. It frames the learning objective as the optimal control of a Markov Decision Process $(\mathcal{S}, \mathcal{A}, P, R_r, \gamma)$ with measurable state space $\mathcal{S}$, discrete actions $\mathcal{A}$, unknown rewards $R_r \in \mathbb{R}^{\mathcal{S} \times \mathcal{A}}$, and unknown dynamics $P \in \mathcal{M}(\mathcal{S})^{\mathcal{S} \times \mathcal{A}}$, where $\mathcal{M}(\mathcal{X})$ denotes the probability measures over a set $\mathcal{X}$. Formally, we seek a policy $\pi \in \mathcal{M}(\mathcal{A})^{\mathcal{S}}$ that maximises in expectation the $\gamma$-discounted return of rewards $G_r^\pi = \sum_{t=0}^{\infty} \gamma^t R_r(s_t, a_t)$.

However, this modelling assumption comes at a price: no control is given over the spread of the performance distribution (Dann *et al.*, 2019). In many critical real-world applications where failures may turn out very costly, this is an issue as most decision-makers would rather give away some amount of expected optimality to increase the performances in the lower-tail of the distribution. This

---

[*]Both authors contributed equally.

[†]Univ. Lille, CNRS, Centrale Lille, INRIA UMR 9189 - CRIStAL, Lille, France

[3]Videos and code are available at https://budgeted-rl.github.io/.

has led to the development of several risk-averse variants where the optimisation criteria include other statistics of the performance, such as the worst-case realisation (Iyengar, 2005; Nilim and El Ghaoui, 2005; Wiesemann *et al.*, 2013), the variance-penalised expectation (García and Fernández, 2015; Tamar *et al.*, 2012), the Value-At-Risk (VaR) (Mausser and Rosen, 2003; Luenberger, 2013), or the Conditional Value-At-Risk (CVaR) (Chow *et al.*, 2015, 2018).

Reinforcement Learning also assumes that the performance can be described by a single reward function $R_r$. Conversely, real problems typically involve many aspects, some of which can be contradictory (Liu *et al.*, 2014). For instance, a self-driving car needs to balance between progressing quickly on the road and avoiding collisions. When aggregating several objectives in a single scalar signal, as often in Multi-Objectives RL (Roijers *et al.*, 2013), no control is given over their relative ratios, as high rewards can compensate high penalties. For instance, if a weighted sum is used to balance velocity $v$ and crashes $c$, then for any given choice of weights $\omega$ the optimality equation $\omega_v \mathbb{E}[\sum \gamma^t v_t] + \omega_a \mathbb{E}[\sum \gamma^t c_t] = G_r^* = \max_\pi G_r^\pi$ is the equation of a line in $(\mathbb{E}[\sum \gamma^t v_t], \mathbb{E}[\sum \gamma^t c_t])$, and the automotive company cannot control where its optimal policy $\pi^*$ lies on that line.

Both of these concerns can be addressed in the *Constrained Markov Decision Process* (CMDP) setting (Beutler and Ross, 1985; Altman, 1999). In this multi-objective formulation, task completion and safety are considered separately. We equip the MDP with a cost signal $R_c \in \mathbb{R}^{\mathcal{S} \times \mathcal{A}}$ and a cost budget $\beta \in \mathbb{R}$. Similarly to $G_r^\pi$, we define the return of costs $G_c^\pi = \sum_{t=0}^{\infty} \gamma^t R_c(s_t, a_t)$ and the new cost-constrained objective:

$$\max_{\pi \in \mathcal{M}(\mathcal{A})^{\mathcal{S}}} \mathbb{E}[G_r^\pi | s_0 = s] \quad \text{s.t.} \quad \mathbb{E}[G_c^\pi | s_0 = s] \leq \beta \tag{1}$$

This constrained framework allows for better control of the performance-safety tradeoff. However, it suffers from a major limitation: the budget has to be chosen before training, and cannot be changed afterwards.

To address this concern, the *Budgeted Markov Decision Process* (BMDP) was introduced in (Boutilier and Lu, 2016) as an extension of CMDPs to enable the online control over the budget $\beta$ within an interval $\mathcal{B} \subset \mathbb{R}$ of admissible budgets. Instead of fixing the budget prior to training, the objective is now to find a generic optimal policy $\pi^*$ that takes $\beta$ as input so as to solve the corresponding CMDP (Eq. (1)) for all $\beta \in \mathcal{B}$. This gives the system designer the ability to move the optimal policy $\pi^*$ in real-time along the Pareto-optimal curve of the different reward-cost trade-offs.

Our first contribution is to re-frame the original BMDP formulation in the context of continuous states and infinite discounted horizon. We then propose a novel Budgeted Bellman Optimality Operator and prove the optimal value function to be a fixed point of this operator. Second, we use this operator in `BFTQ`, a batch Reinforcement Learning algorithm, for solving BMDPs online by interaction with an environment, through function approximation and a tailored exploration procedure. Third, we scale this algorithm to large problems by providing an efficient implementation of the Budgeted Bellman Optimality Operator based on convex programming, a risk-sensitive exploration procedure, and by leveraging tools from Deep Reinforcement Learning such as Deep Neural Networks and synchronous parallel computing. Finally, we validate our approach in two environments that display a clear trade-off between rewards and costs: a spoken dialogue system and a problem of behaviour planning for autonomous driving. The proofs of our main results are provided in Appendix A.

## 2 Budgeted Dynamic Programming

We work in the space of budgeted policies, where a policy $\pi$ both depends on the current budget $\beta$ and also outputs a next budget $\beta_a$. Hence, the budget $\beta$ is neither fixed nor constant as in the CMDP setting but instead evolves as part of the dynamics.

We cast the BMDP problem as a multi-objective MDP problem (Roijers *et al.*, 2013) by considering *augmented* state and action spaces $\overline{\mathcal{S}} = \mathcal{S} \times \mathcal{B}$ and $\overline{\mathcal{A}} = \mathcal{A} \times \mathcal{B}$, and equip them with the augmented dynamics $\overline{P} \in \mathcal{M}(\overline{\mathcal{S}})^{\overline{\mathcal{S}} \times \overline{\mathcal{A}}}$ defined as:

$$\overline{P}\left(\overline{s}' \mid \overline{s}, \overline{a}\right) = \overline{P}\left((s', \beta') \mid (s, \beta), (a, \beta_a)\right) \overset{\text{def}}{=} P(s'|s, a)\delta(\beta' - \beta_a), \tag{2}$$

where $\delta$ is the Dirac indicator distribution.

In other words, in these augmented dynamics, the output budget $\beta_a$ returned at time $t$ by a budgeted policy $\pi \in \Pi = \mathcal{M}(\overline{\mathcal{A}})^{\overline{\mathcal{S}}}$ will be used to condition the policy at the next timestep $t + 1$.

We stack the rewards and cost functions in a single *vectorial* signal $R \in (\mathbb{R}^2)^{\overline{\mathcal{S}} \times \overline{\mathcal{A}}}$. Given an augmented transition $(\overline{s}, \overline{a}) = ((s, \beta), (a, \beta_a))$, we define:

$$R(\overline{s}, \overline{a}) \stackrel{\text{def}}{=} \begin{bmatrix} R_r(s, a) \\ R_c(s, a) \end{bmatrix} \in \mathbb{R}^2. \tag{3}$$

Likewise, the return $G^\pi = (G_r^\pi, G_c^\pi)$ of a budgeted policy $\pi \in \Pi$ refers to: $G^\pi \stackrel{\text{def}}{=} \sum_{t=0}^{\infty} \gamma^t R(\overline{s}_t, \overline{a}_t)$, and the value functions $V^\pi, Q^\pi$ of a budgeted policy $\pi \in \Pi$ are defined as:

$$V^\pi(\overline{s}) = (V_r^\pi, V_c^\pi) \stackrel{\text{def}}{=} \mathbb{E}\left[G^\pi \mid \overline{s}_0 = \overline{s}\right] \qquad Q^\pi(\overline{s}, \overline{a}) = (Q_r^\pi, Q_c^\pi) \stackrel{\text{def}}{=} \mathbb{E}\left[G^\pi \mid \overline{s}_0 = \overline{s}, \overline{a}_0 = \overline{a}\right]. \tag{4}$$

We restrict $\overline{\mathcal{S}}$ to feasible budgets only: $\overline{\mathcal{S}}_f \stackrel{\text{def}}{=} \{(s, \beta) \in \overline{\mathcal{S}} : \exists \pi \in \Pi, V_c^\pi(s) \leq \beta\}$ that we assume is non-empty for the BMDP to admit a solution. We still write $\overline{\mathcal{S}}$ in place of $\overline{\mathcal{S}}_f$ for brevity of notations.

**Proposition 1** (Budgeted Bellman Expectation). *The value functions $V^\pi$ and $Q^\pi$ verify:*

$$V^\pi(\overline{s}) = \sum_{\overline{a} \in \overline{\mathcal{A}}} \pi(\overline{a}|\overline{s}) Q^\pi(\overline{s}, \overline{a}) \qquad Q^\pi(\overline{s}, \overline{a}) = R(\overline{s}, \overline{a}) + \gamma \sum_{\overline{s}' \in \overline{\mathcal{S}}} \overline{P}(\overline{s}' \mid \overline{s}, \overline{a}) V^\pi(\overline{s}') \tag{5}$$

*Moreover, consider the Budgeted Bellman Expectation operator $\mathcal{T}^\pi$: $\forall Q \in (\mathbb{R}^2)^{\overline{\mathcal{S}} \overline{\mathcal{A}}}, \overline{s} \in \overline{\mathcal{S}}, \overline{a} \in \overline{\mathcal{A}}$,*

$$\mathcal{T}^\pi Q(\overline{s}, \overline{a}) \stackrel{\text{def}}{=} R(\overline{s}, \overline{a}) + \gamma \sum_{\overline{s}' \in \overline{\mathcal{S}}} \sum_{\overline{a}' \in \overline{\mathcal{A}}} \overline{P}(\overline{s}'|\overline{s}, \overline{a}) \pi(\overline{a}'|\overline{s}') Q(\overline{s}', \overline{a}') \tag{6}$$

*Then $\mathcal{T}^\pi$ is a $\gamma$-contraction and $Q^\pi$ is its unique fixed point.*

**Definition 1** (Budgeted Optimality). *We now come to the definition of budgeted optimality. We want an optimal budgeted policy to: (i) respect the cost budget $\beta$, (ii) maximise the $\gamma$-discounted return of rewards $G_r$, (iii) in case of tie, minimise the $\gamma$-discounted return of costs $G_c$. To that end, we define for all $\overline{s} \in \overline{\mathcal{S}}$:*

  *(i) Admissible policies $\Pi_a$:*

$$\Pi_a(\overline{s}) \stackrel{\text{def}}{=} \{\pi \in \Pi : V_c^\pi(\overline{s}) \leq \beta\} \text{ where } \overline{s} = (s, \beta) \tag{7}$$

  *(ii) Optimal value function for rewards $V_r^*$ and candidate policies $\Pi_r$:*

$$V_r^*(\overline{s}) \stackrel{\text{def}}{=} \max_{\pi \in \Pi_a(\overline{s})} V_r^\pi(\overline{s}) \qquad\qquad \Pi_r(\overline{s}) \stackrel{\text{def}}{=} \arg\max_{\pi \in \Pi_a(\overline{s})} V_r^\pi(\overline{s}) \tag{8}$$

  *(iii) Optimal value function for costs $V_c^*$ and optimal policies $\Pi^*$:*

$$V_c^*(\overline{s}) \stackrel{\text{def}}{=} \min_{\pi \in \Pi_r(\overline{s})} V_c^\pi(\overline{s}), \qquad\qquad \Pi^*(\overline{s}) \stackrel{\text{def}}{=} \arg\min_{\pi \in \Pi_r(\overline{s})} V_c^\pi(\overline{s}) \tag{9}$$

*We define the budgeted action-value function $Q^*$ similarly:*

$$Q_r^*(\overline{s}, \overline{a}) \stackrel{\text{def}}{=} \max_{\pi \in \Pi_a(\overline{s})} Q_r^\pi(\overline{s}, \overline{a}) \qquad\qquad Q_c^*(\overline{s}, \overline{a}) \stackrel{\text{def}}{=} \min_{\pi \in \Pi_r(\overline{s})} Q_c^\pi(\overline{s}, \overline{a}) \tag{10}$$

*and denote $V^* = (V_r^*, V_c^*)$, $Q^* = (Q_r^*, Q_c^*)$.*

**Theorem 1** (Budgeted Bellman Optimality). *The optimal budgeted action-value function $Q^*$ verifies:*

$$Q^*(\overline{s}, \overline{a}) = \mathcal{T}Q^*(\overline{s}, \overline{a}) \stackrel{\text{def}}{=} R(\overline{s}, \overline{a}) + \gamma \sum_{\overline{s}' \in \overline{\mathcal{S}}} \overline{P}(\overline{s}'|\overline{s}, \overline{a}) \sum_{\overline{a}' \in \overline{\mathcal{A}}} \pi_{greedy}(\overline{a}'|\overline{s}'; Q^*) Q^*(\overline{s}', \overline{a}'), \tag{11}$$

*where the greedy policy $\pi_{greedy}$ is defined by: $\forall \overline{s} = (s, \beta) \in \overline{\mathcal{S}}, \overline{a} \in \overline{\mathcal{A}}, \forall Q \in (\mathbb{R}^2)^{\overline{\mathcal{A}} \times \overline{\mathcal{S}}}$,*

$$\pi_{greedy}(\overline{a}|\overline{s}; Q) \in \arg\min_{\rho \in \Pi_r^Q} \mathbb{E}_{\overline{a} \sim \rho} Q_c(\overline{s}, \overline{a}), \tag{12a}$$

$$where \quad \Pi_r^Q \stackrel{\text{def}}{=} \arg\max_{\rho \in \mathcal{M}(\overline{\mathcal{A}})} \mathbb{E}_{\overline{a} \sim \rho} Q_r(\overline{s}, \overline{a}) \tag{12b}$$

$$s.t. \quad \mathbb{E}_{\overline{a} \sim \rho} Q_c(\overline{s}, \overline{a}) \leq \beta. \tag{12c}$$

**Remark 1** (Appearance of the greedy policy). *In classical Reinforcement Learning, the greedy policy takes a simple form $\pi_{greedy}(s;Q^*) = \arg\max_{a \in \mathcal{A}} Q^*(s,a)$, and the term $\pi_{greedy}(a'|s';Q^*)Q^*(s',a')$ in (11) conveniently simplifies to $\max_{a' \in \mathcal{A}} Q^*(s',a')$. Unfortunately, in a budgeted setting the greedy policy requires solving the nested constrained optimisation program (12) at each state and budget in order to apply this Budgeted Bellman Optimality operator.*

**Proposition 2** (Optimality of the greedy policy). *The greedy policy $\pi_{greedy}(\cdot\,;Q^*)$ is uniformly optimal: $\forall \overline{s} \in \overline{\mathcal{S}}$, $\pi_{greedy}(\cdot\,;Q^*) \in \Pi^*(\overline{s})$. In particular, $V^{\pi_{greedy}(\cdot;Q^*)} = V^*$ and $Q^{\pi_{greedy}(\cdot;Q^*)} = Q^*$.*

**Budgeted Value Iteration**  The Budgeted Bellman Optimality equation is a fixed-point equation, which motivates the introduction of a fixed-point iteration procedure. We introduce Algorithm 1, a Dynamic Programming algorithm for solving known BMDPs. If it were to converge to a unique fixed point, this algorithm would provide a way to compute $Q^*$ and recover the associated optimal budgeted policy $\pi_{greedy}(\cdot\,;Q^*)$.

**Theorem 2** (Non-contractivity of $\mathcal{T}$). *For any BMDP $(\mathcal{S}, \mathcal{A}, P, R_r, R_c, \gamma)$ with $|\mathcal{A}| \geq 2$, $\mathcal{T}$ is not a contraction. Precisely: $\forall \varepsilon > 0, \exists Q^1, Q^2 \in (\mathbb{R}^2)^{\overline{\mathcal{S}\mathcal{A}}} : \|\mathcal{T}Q^1 - \mathcal{T}Q^2\|_\infty \geq \frac{1}{\varepsilon}\|Q^1 - Q^2\|_\infty$.*

Unfortunately, as $\mathcal{T}$ is not a contraction, we can guarantee neither the convergence of Algorithm 1 nor the unicity of its fixed points. Despite those theoretical limitations, we empirically observed the convergence to a fixed point in our experiments (Section 5). We conjecture a possible explanation:

**Theorem 3** (Contractivity of $\mathcal{T}$ on smooth $Q$-functions). *The operator $\mathcal{T}$ is a contraction when restricted to the subset $\mathcal{L}_\gamma$ of $Q$-functions such that "$Q_r$ is Lipschitz with respect to $Q_c$":*

$$\mathcal{L}_\gamma = \left\{ \begin{array}{l} Q \in (\mathbb{R}^2)^{\overline{\mathcal{S}\mathcal{A}}} \text{ s.t. } \exists L < \frac{1}{\gamma} - 1 : \forall \overline{s} \in \overline{\mathcal{S}}, \overline{a}_1, \overline{a}_2 \in \overline{\mathcal{A}}, \\ |Q_r(\overline{s}, \overline{a}_1) - Q_r(\overline{s}, \overline{a}_2)| \leq L|Q_c(\overline{s}, \overline{a}_1) - Q_c(\overline{s}, \overline{a}_2)| \end{array} \right\} \quad (13)$$

Thus, we expect that Algorithm 1 is likely to converge when $Q^*$ is smooth, but could diverge if the slope of $Q^*$ is too high. $L^2$-regularisation can be used to encourage smoothness and mitigate risk of divergence.

## 3    Budgeted Reinforcement Learning

In this section, we consider BMDPs with unknown parameters that must be solved by interaction with an environment.

### 3.1    Budgeted Fitted-Q

When the BMDP is unknown, we need to adapt Algorithm 1 to work with a batch of samples $\mathcal{D} = \{(\overline{s}_i, \overline{a}_i, r_i, \overline{s}'_i)\}_{i \in [1,N]}$ collected by interaction with the environment. Applying $\mathcal{T}$ in (11) would require computing an expectation $\mathbb{E}_{\overline{s}' \sim \overline{P}}$ over next states $\overline{s}'$ and hence an access to the model $\overline{P}$. We instead use $\hat{\mathcal{T}}$, a sampling operator, in which this expectation is replaced by:

$$\hat{\mathcal{T}}Q(\overline{s}, \overline{a}, r, \overline{s}') \stackrel{\text{def}}{=} r + \gamma \sum_{\overline{a}' \in \overline{\mathcal{A}}} \pi_{greedy}(\overline{a}'|\overline{s}';Q)Q(\overline{s}', \overline{a}').$$

We introduce in Algorithm 2 the *Budgeted-Fitted-Q* (BFTQ) algorithm, an extension of the *Fitted-Q* (FTQ) algorithm (Ernst *et al.*, 2005; Riedmiller, 2005) adapted to solve unknown BMDPs. Because we work with continuous state space $\mathcal{S}$ and budget space $\mathcal{B}$, we need to employ function-approximation in order to generalise to nearby states and budgets. Precisely, given a parametrized model $Q_\theta$, we seek to minimise a regression loss $\mathcal{L}(Q_\theta, Q_{\text{target}}; \mathcal{D}) = \sum_{\mathcal{D}} \|Q_\theta(\overline{s}, \overline{a}) - Q_{\text{target}}(\overline{s}, \overline{a}, r, \overline{s}')\|_2^2$. Any model can be used, such as linear models, regression trees, or neural networks.

---

**Algorithm 1:** Budgeted Value Iteration

**Data:** $P, R_r, R_c$
**Result:** $Q^*$
1  $Q_0 \leftarrow 0$
2  **repeat**
3  $\quad \big| \quad Q_{k+1} \leftarrow \mathcal{T}Q_k$
4  **until** *convergence*

---

**Algorithm 2:** Budgeted Fitted-Q

**Data:** $\mathcal{D}$
**Result:** $Q^*$
1  $Q_{\theta_0} \leftarrow 0$
2  **repeat**
3  $\quad \big| \quad \theta_{k+1} \leftarrow \arg\min_\theta \mathcal{L}(Q_\theta, \hat{\mathcal{T}}Q_{\theta_k}; \mathcal{D})$
4  **until** *convergence*

---

## 3.2 Risk-sensitive exploration

In order to run Algorithm 2, we must first gather a batch of samples $\mathcal{D}$. The following strategy is motivated by the intuition that a wide variety of risk levels needs to be experienced during training, which can be achieved by enforcing the risk constraints during data collection. Ideally we would need samples from the asymptotic state-budget distribution $\lim_{t\to\infty} \mathbb{P}(\bar{s}_t)$ induced by an optimal policy $\pi^*$ given an initial distribution $\mathbb{P}(\bar{s}_0)$, but as we are actually building this policy, it is not possible. Following the same idea of $\varepsilon$-greedy exploration for FTQ (Ernst *et al.*, 2005; Riedmiller, 2005), we introduce an algorithm for risk-sensitive exploration. We follow an exploration policy: a mixture between a random budgeted policy $\pi_{\text{rand}}$ and the current greedy policy $\pi_{\text{greedy}}$. The batch $\mathcal{D}$ is split into several mini-batches generated sequentially, and $\pi_{\text{greedy}}$ is updated by running Algorithm 2 on $\mathcal{D}$ upon mini-batch completion. $\pi_{\text{rand}}$ should only pick augmented actions that are admissible candidates for $\pi_{\text{greedy}}$. To that extent $\pi_{\text{rand}}$ is designed to obtain trajectories that only explore feasible budgets: we impose that the joint distribution $\mathbb{P}(a, \beta_a | s, \beta)$ verifies $\mathbb{E}[\beta_a] \leq \beta$. This condition defines a probability simplex $\Delta_{\overline{\mathcal{A}}}$ from which we sample uniformly. Finally, when interacting with an environment the initial state $s_0$ is usually sampled from a starting distribution $\mathbb{P}(s_0)$. In the budgeted setting, we also need to sample the initial budget $\beta_0$. Importantly, we pick a uniform distribution $\mathbb{P}(\beta_0) = \mathcal{U}(\mathcal{B})$ so that the entire range of risk-level is explored, and not only reward-seeking behaviours as would be the case with a traditional risk-neutral $\varepsilon$-greedy strategy. The pseudo-code of our exploration procedure is shown in Algorithm 3.

---

**Algorithm 3:** Risk-sensitive exploration

**Data:** An environment, a BFTQ solver, $W$ CPU workers
**Result:** A batch of transitions $\mathcal{D}$

1   $\mathcal{D} \leftarrow \emptyset$
2   **for** *each intermediate batch* **do**
3      split episodes between $W$ workers
4      **for** *each episode in batch* **do**        `// run this loop on each worker in parallel`
5          sample initial budget $\beta \sim \mathcal{U}(\mathcal{B})$.
6          **while** *episode not done* **do**
7             update $\varepsilon$ from schedule.
8             sample $z \sim \mathcal{U}([0, 1])$.
9             **if** $z < \varepsilon$ **then** sample $(a, \beta_a) \sim \mathcal{U}(\Delta_{\mathcal{AB}})$.        `// Explore`
10            **else** sample $(a, \beta_a) \sim \pi_{\text{greedy}}(a, \beta_a | s, \beta; Q^*)$.        `// Exploit`
11             append transition $(s, \beta, a, \beta_a, R, C, s')$ to batch $\mathcal{D}$.
12             step episode budget $\beta \leftarrow \beta_a$
13          **end**
14      **end**
15      $\pi_{\text{greedy}}(\cdot \sim; Q^*) \leftarrow \text{BFTQ}(\mathcal{D})$.
16   **end**
17   **return** *the batch of transitions* $\mathcal{D}$

---

## 4 A Scalable Implementation

In this section, we introduce an implementation of the BFTQ algorithm designed to operate efficiently and handle large batches of experiences $\mathcal{D}$.

### 4.1 How to compute the greedy policy?

As stated in Remark 1, computing the greedy policy $\pi_{\text{greedy}}$ in (11) is not trivial since it requires solving the nested constrained optimisation program (12). However, it can be solved efficiently by exploiting the *structure* of the set of solutions with respect to $\beta$, that is, concave and increasing.

**Proposition 3** (Equality of $\pi_{\text{greedy}}$ and $\pi_{\text{hull}}$)**.** *Algorithm 1 and Algorithm 2 can be run by replacing $\pi_{greedy}$ in the equation* (11) *of* $\mathcal{T}$ *with $\pi_{hull}$ as described in Algorithm 4.*

$$\pi_{greedy}(\overline{a}|\overline{s}; Q) = \pi_{hull}(\overline{a}|\overline{s}; Q)$$

**Algorithm 4:** Convex hull policy $\pi_{\text{hull}}(\overline{a}|\overline{s}; Q)$

---

**Data:** $\overline{s} = (s, \beta), Q$

1   $Q^+ \leftarrow \{Q_c > \min\{Q_c(\overline{s}, \overline{a}) \text{ s.t. } \overline{a} \in \arg\max_{\overline{a}} Q_r(\overline{s}, \overline{a})\}\}$     `// dominated points`

2   $\mathcal{F} \leftarrow$ top frontier of `convex_hull`$(Q(\overline{s}, \overline{\mathcal{A}}) \setminus Q^+)$     `// candidate mixtures`

3   $\mathcal{F}_Q \leftarrow \mathcal{F} \cap Q(\overline{s}, \overline{\mathcal{A}})$

4   **for** *points* $q = Q(\overline{s}, \overline{a}) \in \mathcal{F}_Q$ *in clockwise order* **do**

5      **if** *find two successive points* $((q_c^1, q_r^1), (q_c^2, q_r^2))$ *of* $\mathcal{F}_Q$ *such that* $q_c^1 \leq \beta < q_c^2$ **then**

6         $p \leftarrow (\beta - q_c^1)/(q_c^2 - q_c^1)$

7         **return** the mixture $(1 - p)\delta(\overline{a} - \overline{a}^1) + p\delta(\overline{a} - \overline{a}^2)$

8   **end**

9   **return** $\delta(\overline{a} - \arg\max_{\overline{a}} Q_r(\overline{s}, \overline{a}))$          `// budget` $\beta$ `always respected`

---

The computation of $\pi_{\text{hull}}$ in Algorithm 4 is illustrated in Figure 1: first we get rid of dominated points. Then we compute the top frontier of the convex hull of the $Q$-function. Next, we find the two closest augmented actions $\overline{a}_1$ and $\overline{a}_2$ with cost-value $Q_c$ surrounding $\beta$: $Q_c(\overline{s}, \overline{a}_1) \leq \beta < Q_c(\overline{s}, \overline{a}_2)$. Finally, we mix the two actions such that the expected spent budget is equal to $\beta$. Because of the concavity of the convex hull top frontier, any other combination of augmented actions would lead to a lower expected reward $Q_r$.

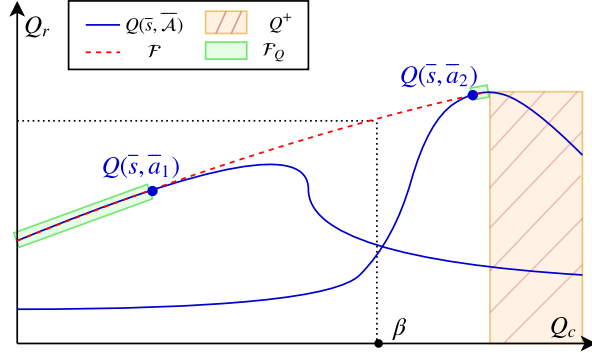

Figure 1: Representation of $\pi_{\text{hull}}$. When the budget lies between $Q(\overline{s}, \overline{a}_1)$ and $Q(\overline{s}, \overline{a}_2)$, two points of the top frontier of the convex hull, then the policy is a mixture of these two points.

## 4.2 Function approximation

Neural networks are well suited to model $Q$-functions in Reinforcement Learning algorithms (Riedmiller, 2005; Mnih *et al.*, 2015). We approximate $Q = (Q_r, Q_c)$ using one single neural network. Thus, the two components are jointly optimised which accelerates convergence and fosters learning of useful shared representations. Moreover, as in (Mnih *et al.*, 2015) we are dealing with a finite (categorical) action space $\mathcal{A}$, instead of including the action in the input we add the output of the $Q$-function for each action to the last layer. Again, it provides a faster convergence toward useful shared representations and it only requires one forward pass to evaluate all action values. Finally, beside the state $s$ there is one more input to a budgeted $Q$-function: the budget $\beta_a$. This budget is a scalar value whereas the state $s$ is a vector of potentially large size. To avoid a weak influence of $\beta$ compared to $s$ in the prediction, we include an additional encoder for the budget, whose width and depth may depend on the application. A straightforward choice is a single layer with the same width as the state. The overall architecture is shown in Figure 7 in Appendix B.

## 4.3 Parallel computing

In a simulated environment, a first process that can be distributed is the collection of samples in the exploration procedure of Algorithm 3, as $\pi_{\text{greedy}}$ stays constant within each mini-batch which avoids the need of synchronisation between workers. Second, the main bottleneck of BFTQ is the computation of the target $\mathcal{T}Q$. Indeed, when computing $\pi_{\text{hull}}$ we must perform at each epoch a Graham-scan of complexity $\mathcal{O}(|\mathcal{A}||\widetilde{\mathcal{B}}| \log |\mathcal{A}\widetilde{\mathcal{B}}|)$ per sample in $\mathcal{D}$ to compute the convex hulls of $Q$ (where $\widetilde{\mathcal{B}}$ is a finite discretisation of $\mathcal{B}$). The resulting total time-complexity is $\mathcal{O}(\frac{|\mathcal{D}||\mathcal{A}||\widetilde{\mathcal{B}}|}{1-\gamma} \log |\mathcal{A}||\widetilde{\mathcal{B}}|)$. This operation can easily be distributed over several CPUs provided that we first evaluate the model $Q(s', \mathcal{A}\widetilde{\mathcal{B}})$ for each sample $s' \in \mathcal{D}$, which can be done in a single forward pass. By using multiprocessing in the computations of $\pi_{\text{hull}}$, we enjoy a linear speedup. The full description of our scalable implementation of BFTQ is recalled in Algorithm 5 in Appendix B.

# 5 Experiments

There are two hypotheses we want to validate.

**Exploration strategies**   We claimed in Section 3.2 that a risk-sensitive exploration was required in the setting of BMDPs. We test this hypotheses by confronting our strategy to a classical risk-neutral strategy. The latter is chosen to be a $\varepsilon$-greedy policy slowly transitioning from a random to a greedy policy[4] that aims to maximise $\mathbb{E}_\pi \, G_r^\pi$ regardless of $\mathbb{E}_\pi \, G_c^\pi$. The quality of the resulting batches $\mathcal{D}$ is assessed by training a BFTQ policy and comparing the resulting performance.

**Budgeted algorithms**   We compare our scalable BFTQ algorithm described in Section 4 to an FTQ($\lambda$) baseline. This baseline consists in approximating the BMDP by a finite set of CMDPs problems. We solve each of these CMDP using the standard technique of Lagrangian Relaxation: the cost constraint is converted to a soft penalty weighted by a Lagrangian multiplier $\lambda$ in a surrogate reward function: $\max_\pi \mathbb{E}_\pi [G_r^\pi - \lambda G_c^\pi]$. The resulting MDP can be solved by any RL algorithm, and we chose FTQ for being closest to BFTQ. In our experiments, a single training of BFTQ corresponds to 10 trainings of FTQ($\lambda$) policies. Each run was repeated $N_{\text{seeds}}$ times. Parameters of the algorithms can be found in Appendix D.3.1

## 5.1 Environments

We evaluate our method on three different environments involving reward-cost trade-offs. Their parameters can be found in Appendix D.3.2

**Corridors**   This simple environment is only meant to highlight clearly the specificity of exploration in a budgeted setting. It is a continuous gridworld with Gaussian perturbations, consisting in a maze composed of two corridors: a risky one with high rewards and costs, and a safe one with low rewards and no cost. In both corridors the outermost cell is the one yielding the most reward, which motivates a deep exploration.

**Spoken dialogue system**   Our second application is a dialogue-based slot-filling simulation that has already benefited from batch RL optimisation in the past (Li *et al.*, 2009; Chandramohan *et al.*, 2010; Pietquin *et al.*, 2011). The system fills in a form of slot-values by interacting a user through speech, before sending them a response. For example, in a restaurant reservation domain, it may ask for three slots: the area of the restaurant, the price-range and the food type. The user could respectively provide those three slot-values : Cambridge, Cheap and Indian-food. In this application, we do not focus on how to extract such information from the user utterances, we rather focus on decision-making for filling in the form. To that end, the system can choose among a set of generic actions. As in (Carrara *et al.*, 2018), there are two ways of asking for a slot value: a slot value can be either be provided with an utterance, which may cause speech recognition errors with some probability, or by requiring the user to fill-in the slots by using a numeric pad. In this case, there are no recognition errors but a counterpart risk of hang-up: we assume that manually filling a key-value form is time-consuming and annoying. The environment yields a reward if all slots are filled without errors, and a constraint if the user hang-ups. Thus, there is a clear trade-off between using utterances and potentially committing a mistake, or using the numeric pad and risking a premature hang-up.

**Autonomous driving**   In our third application, we use the highway-env environment (Leurent, 2018) for simulated highway driving and behavioural decision-making. We define a task that displays a clear trade-off between safety and efficiency. The agent controls a vehicle with a finite set of manoeuvres implemented by low-lever controllers: $\mathcal{A}$ = {no-op, right-lane, left-lane, faster, slower}. It is driving on a two-lane road populated with other traffic participants: the vehicles in front of the agent drive slowly, and there are incoming vehicles on the opposite lane. Their behaviours are randomised, which introduces some uncertainty with respect to their possible future trajectories. The task consists in driving as fast as possible, which is modelled by a reward proportional to the velocity: $R_r(s_t, a_t) \propto v_t$. This motivates the agent to try and overtake its preceding vehicles by driving fast on the opposite lane. This optimal but overly aggressive behaviour can be tempered through a cost function that embodies a safety objective: $R_c(s_t, a_t)$ is set to $1/H$ whenever the ego-vehicle is

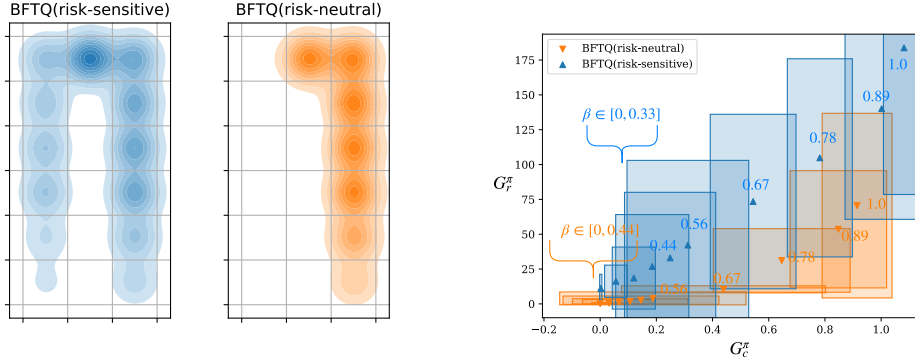

Figure 2: Density of explored states (left) and corresponding policy performances (right) of two exploration strategies in the corridors environment.

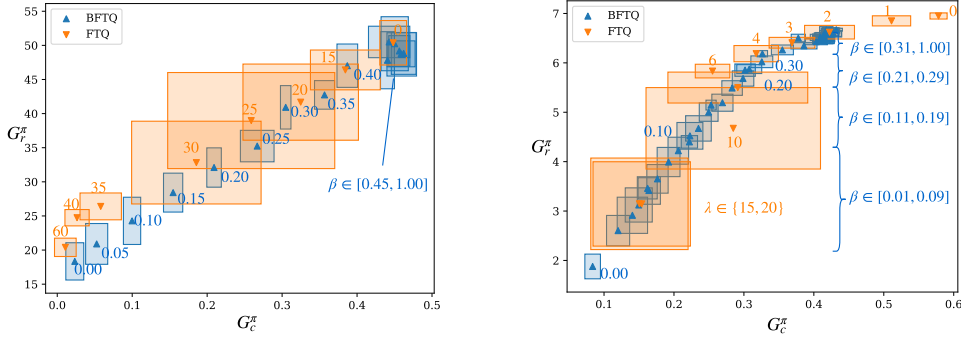

Figure 3: Performance comparison of FTQ($\lambda$) and BFTQ on slot-filling (left) and highway-env(right)

driving on the opposite lane, where $H$ is the episode horizon. Thus, the constrained signal $G_c^\pi$ is the maximum proportion of time that the agent is allowed to drive on the wrong side of the road.

## 5.2 Results

In the following figures, each patch represents the mean and 95% confidence interval over $N_{\text{seeds}}$ seeds of the means of $(G_r^\pi, G_c^\pi)$ over $N_{\text{trajs}}$ trajectories. That way, we display the variation related to learning (and batches) rather than the variation in the execution of the policies.

We first bring to light the role of risk-sensitive exploration in the corridors environment: Figure 2 shows the set of trajectories collected by each exploration strategy. and the resulting performance of a budgeted policy trained on each batch. The trajectories (orange) in the risk-neutral batch are concentrated along the risky corridor (right) and ignore the safe corridor (left), which results in bad performances in the low-risk regime. Conversely, trajectories in the risk-sensitive batch (blue) are well distributed among both corridors and the corresponding budgeted policy achieves good performance across the whole spectrum of risk budgets.

In a second experiment displayed in Figure 3, we compare the performance of FTQ($\lambda$) to that of BFTQ in the dialogue and autonomous driving tasks. For each algorithm, we plot the reward-cost trade-off curve. In both cases, BFTQ performs almost as well as FTQ($\lambda$) despite only requiring a single model. All budgets are well-respected on slot-filling, but on highway-env we can observe an underestimation of $Q_c$, since e.g. $\mathbb{E}[G_c|\beta = 0] \simeq 0.1$. This underestimation can be a consequence of two approximations: the use of the sampling operator $\hat{\mathcal{T}}$ instead of the true population operator $\mathcal{T}$, and the use of the neural network function approximation $Q_\theta$ instead of $Q$. Still, BFTQ provides a better control on the expected cost of the policy, than FTQ($\lambda$). In addition, BFTQ behaves more consistently than FTQ($\lambda$) overall, as shown by its lower extra-seed variance.

Additional material such as videos of policy executions is provided in Appendix D.

## 6  Discussion

Algorithm 2 is an algorithm for solving large unknown BMDPs with continuous states. To the best of our knowledge, there is no algorithm in the current literature that combines all those features.

Algorithms have been proposed for CMDPs, which are less flexible sub-problems of the more general BMDP. When the environment parameters $(P, R_r, R_c)$ are known but not tractable, solutions relying on function approximation (Undurti *et al.*, 2011) or approximate linear programming (Poupart *et al.*, 2015) have been proposed. For unknown environments, online algorithms (Geibel and Wysotzki, 2005; Abe and others, 2010; Chow *et al.*, 2018; Achiam *et al.*, 2017) and a batch algorithm (Thomas *et al.*, 2015; Petrik *et al.*, 2016; Laroche and Trichelair, 2019; Le *et al.*, 2019) can solve large unknown CMDPs. Nevertheless, these approaches are limited in that the constraints thresholds are fixed prior to training and cannot be updated in real-time at policy execution to select the desired level of risk.

To our knowledge, there were only two ways of solving a BMDP. The first one is to approximate it with a finite set of CMDPs (e.g. see our `FTQ`($\lambda$) baseline). The solutions of these CMDPs take the form of mixtures between two deterministic policies (Theorem 4.4, Beutler and Ross, 1985). To obtain these policies, one needs to evaluate their expected cost by interacting with the environment[5]. Our solution not only requires one single model but also avoids any supplementary interaction.

The only other existing BMDP algorithm, and closest work to ours, is the Dynamic Programming algorithm proposed by Boutilier and Lu (2016). However, their work was established for finite state spaces only, and their solution relies heavily on this property. For instance, they enumerate and sort the next states $s' \in \mathcal{S}$ by their expected value-by-cost, which could not be performed in a continuous state space $\mathcal{S}$. Moreover, they rely on the knowledge of the model $(P, R_r, R_c)$, and do not address the question of learning from interaction data.

## 7  Conclusion

The BMDP framework is a principled framework for safe decision making under uncertainty, which could be beneficial to the diffusion of Reinforcement Learning in industrial applications. However, BMDPs could so far only be solved in finite state spaces which limits their interest in many use-cases. We extend their definition to continuous states by introducing of a novel Dynamic Programming operator, that we build upon to propose a Reinforcement Learning algorithm. In order to scale to large problems, we provide an efficient implementation that exploits the structure of the value function and leverages tools from Deep Distributed Reinforcement Learning. We show that on two practical tasks our solution performs similarly to a baseline Lagrangian relaxation method while only requiring a single model to train, and relying on an interpretable $\beta$ instead of the tedious tuning of the penalty $\lambda$.

## Acknowledgements

This work has been supported by CPER Nord-Pas de Calais/FEDER DATA Advanced data science and technologies 2015-2020, the French Ministry of Higher Education and Research, INRIA, and the French Agence Nationale de la Recherche (ANR). We thank Guillaume Gautier, Fabrice Clerot, and Xuedong Shang for the helpful discussions and valuable insights.

## Footnotes

[4]We train this greedy policy using FTQ.

[5]More details are provided in Appendix C

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
