[Supplementary Material]

# Appendices

**Outline** This paper gathers all the supplementary material and goes as follows: Appendix A details all the proofs of the main results. Appendix B recalls the full scalable BFTQ algorithm. Appendix C describes a naive alternative to BFTQ based on Lagrangian Relaxation. The Appendix D assembles all the assets for visualising and reproducing the experiments, including visualisations of policy executions, algorithms and environment parameters, and instructions for executing the attached source code. Finally we fill the Machine Learning Reproducibility Checklist and we justify each statement in Appendix E. Please note that all the materials referenced through web links are also available in the zip file of supplementary materials.

## A Proofs of Main Results

### A.1 Proposition 1

*Proof.* Thanks to the introduction of the augmented spaces $\overline{\mathcal{S}}, \overline{\mathcal{A}}$ and dynamics $\overline{P}$, this proof is the same as that in classical multi-objective MDPs.

$$
\begin{aligned}
V^\pi(\overline{s}) &\overset{\text{def}}{=} \mathbb{E}\left[G^\pi \mid \overline{s_0} = \overline{s}\right] \\
&= \sum_{\overline{a} \in \overline{\mathcal{A}}} \mathbb{P}\left(\overline{a}_0 = \overline{a} \mid \overline{s_0} = \overline{s}\right) \mathbb{E}\left[G^\pi \mid \overline{s_0} = \overline{s}, \overline{a}_0 = \overline{a}\right] \\
&= \sum_{\overline{a} \in \overline{\mathcal{A}}} \pi(\overline{a}|\overline{s}) Q^\pi(\overline{s}, \overline{a})
\end{aligned}
$$

$$
\begin{aligned}
Q^\pi(\overline{s}, \overline{a}) &\overset{\text{def}}{=} \mathbb{E}\left[\sum_{t=0}^\infty \gamma^t R(\overline{s}_t, \overline{a}_t) \,\middle|\, \overline{s_0} = \overline{s}, \overline{a}_0 = \overline{a}\right] \\
&= R(\overline{s}, \overline{a}) + \sum_{\overline{s}' \in \overline{\mathcal{S}}} \mathbb{P}\left(\overline{s}_1 = \overline{s}' \mid \overline{s_0} = \overline{s}, \overline{a_0} = \overline{a}\right) \cdot \mathbb{E}\left[\sum_{t=1}^\infty \gamma^t R(\overline{s}_t, \overline{a}_t) \,\middle|\, \overline{s_1} = \overline{s}'\right] \\
&= R(\overline{s}, \overline{a}) + \gamma \sum_{\overline{s}' \in \overline{\mathcal{S}}} \overline{P}\left(\overline{s}' \mid \overline{s}, \overline{a}\right) \mathbb{E}\left[\sum_{t=0}^\infty \gamma^t R(\overline{s}_t, \overline{a}_t) \,\middle|\, \overline{s_0} = \overline{s}'\right] \\
&= R(\overline{s}, \overline{a}) + \gamma \sum_{\overline{s}' \in \overline{\mathcal{S}}} \overline{P}\left(\overline{s}' \mid \overline{s}, \overline{a}\right) V^\pi(\overline{s}')
\end{aligned}
$$

**Contraction of $\mathcal{T}^\pi$:** Let $\pi \in \Pi, Q_1, Q_2 \in (\mathbb{R}^2)^{\overline{\mathcal{S}\mathcal{A}}}$.

$$
\begin{aligned}
\forall \overline{s} \in \overline{\mathcal{S}}, \overline{a} \in \overline{\mathcal{A}}, \quad |\mathcal{T}^\pi Q_1(\overline{s}, \overline{a}) - \mathcal{T}^\pi Q_2(\overline{s}, \overline{a})| &= \left| \gamma \mathop{\mathbb{E}}_{\substack{\overline{s}' \sim \overline{P}(\overline{s}'|\overline{s}, \overline{a}) \\ \overline{a}' \sim \pi(\overline{a}'|\overline{s}')}} Q_1(\overline{s}', \overline{a}') - Q_2(\overline{s}', \overline{a}') \right| \\
&\leq \gamma \|Q_1 - Q_2\|_\infty
\end{aligned}
$$

Hence, $\|\mathcal{T}^\pi Q_1 - \mathcal{T}^\pi Q_2\|_\infty \leq \gamma \|Q_1 - Q_2\|_\infty$

According to the Banach fixed point theorem, $\mathcal{T}^\pi$ admits a unique fixed point. It can be easily verified that $Q^\pi$ is indeed this fixed point by combining the two Bellman Expectation equations (5).

□

### A.2 Theorem 1

*Proof.* Let $\overline{s}, \overline{a} \in \overline{\mathcal{A}} \times \overline{\mathcal{S}}$. For this proof, we consider potentially non-stationary policies $\pi = (\rho, \pi')$, with $\rho \in \mathcal{M}(\overline{\mathcal{A}})$, $\pi' \in \mathcal{M}(\overline{\mathcal{A}})^{\mathbb{N}}$. The results will apply to the particular case of stationary optimal policies, when they exist.

$$Q_r^*(\overline{s}, \overline{a}) = \max_{\rho, \pi'} Q_r^{\rho, \pi'}(\overline{s}', \overline{a}') \tag{14}$$

$$= \max_{\rho, \pi'} R_r(s, a) + \gamma \sum_{\overline{s}' \in \overline{\mathcal{S}}} P(\overline{s}'|\overline{s}, \overline{a}) V_r^{\rho, \pi'}(\overline{s}') \tag{15}$$

$$= R_r(s, a) + \gamma \sum_{\overline{s}' \in \overline{\mathcal{S}}} P(\overline{s}'|\overline{s}, \overline{a}) \max_{\rho, \pi'} \sum_{\overline{a}' \in \overline{\mathcal{A}}} \rho(\overline{a}'|\overline{s}') Q_r^{\pi'}(\overline{s}', \overline{a}') \tag{16}$$

$$= R_r(s, a) + \gamma \sum_{\overline{s}' \in \overline{\mathcal{S}}} P(\overline{s}'|\overline{s}, \overline{a}) \max_{\rho} \sum_{\overline{a}' \in \overline{\mathcal{A}}} \rho(\overline{a}'|\overline{s}') \max_{\pi' \in \Pi_a(\overline{s}')} Q_r^{\pi'}(\overline{s}', \overline{a}') \tag{17}$$

$$= R_r(s, a) + \gamma \sum_{\overline{s}' \in \overline{\mathcal{S}}} P(\overline{s}'|\overline{s}, \overline{a}) \max_{\rho} \mathbb{E}_{\overline{a}' \sim \rho} Q_r^*(\overline{s}', \overline{a}') \tag{18}$$

where $\pi = (\rho, \pi') \in \Pi_a(\overline{s})$ and $\pi' \in \Pi_a(\overline{s}')$.

This follows from:

- (14). Definition of $Q^*$.

- (15). Bellman Expectation expansion from Proposition 1.

- (16). Marginalisation on $\overline{a}'$.

- (17). 
  - Trivially $\max_{\pi' \in \Pi_a(\overline{s}')} \sum_{\overline{a}' \in \overline{\mathcal{A}}} \cdot \leq \sum_{\overline{a}' \in \overline{\mathcal{A}}} \max_{\pi' \in \Pi_a(\overline{s})} \cdot$.
  - Let $\overline{\pi} \in \arg\max_{\pi' \in \Pi_a(\overline{s}')} Q_r^{\pi'}(\overline{s}', \overline{a}')$, then:

$$\sum_{\overline{a}' \in \overline{\mathcal{A}}} \rho(\overline{a}'|\overline{s}') \max_{\pi' \in \Pi_a(\overline{s}')} Q_r^{\pi'}(\overline{s}', \overline{a}') = \sum_{\overline{a}' \in \overline{\mathcal{A}}} \rho(\overline{a}'|\overline{s}') Q_r^{\overline{\pi}}(\overline{s}', \overline{a}')$$

$$\leq \max_{\pi' \in \Pi_a(\overline{s}')} \sum_{\overline{a}' \in \overline{A}} \rho(\overline{a}'|\overline{s}') Q_r^{\pi'}(\overline{s}', \overline{a}')$$

- (18). Definition of $Q^*$.

Moreover, the condition $\pi = (\rho, \pi') \in \Pi_a(\overline{s})$ gives

$$\mathbb{E}_{\overline{a}' \sim \rho} Q_c^*(\overline{s}, \overline{a}) = \mathbb{E}_{\overline{a}' \sim \rho} Q_c^{\pi'}(\overline{s}, \overline{a}) = V_c^{\pi}(\overline{s}) \leq \beta$$

Consequently, $\pi_{\text{greedy}}(\cdot; Q^*)$ belongs to the $\arg\max$ of (18), and in particular:

$$Q_r^*(\overline{s}, \overline{a}) = r(\overline{s}, \overline{a}) + \gamma \sum_{\overline{s}' \in \mathcal{S}} P(\overline{s}'|\overline{s}, \overline{a}) \mathbb{E}_{\overline{a}' \sim \pi_{\text{greedy}}(\overline{s}', Q^*)} Q_r^*(\overline{s}', \overline{a}')$$

The same reasoning can be made for $Q_c^*$ by replacing $\max$ operators by $\min$, and $\Pi_a$ by $\Pi_r$. $\qquad \square$

## A.3   Proposition 2

*Proof.* Notice from the definitions of $\mathcal{T}$ and $\mathcal{T}^\pi$ in (11) and (6) that $\mathcal{T}$ and $\mathcal{T}^{\pi_{\text{greedy}}(\cdot; Q^*)}$ coincide on $Q^*$. Moreover, since $Q^* = \mathcal{T}Q^*$ by Theorem 1, we have: $\mathcal{T}^{\pi_{\text{greedy}}(\cdot; Q^*)} Q^* = \mathcal{T}Q^* = Q^*$. Hence, $Q^*$ is a fixed point of $\mathcal{T}^{\pi_{\text{greedy}}(\cdot; Q^*)}$, and by Proposition 1 it must be equal to $Q^{\pi_{\text{greedy}}(\cdot; Q^*)}$

To show the same result for $V^*$, notice that

$$V^{\pi_{\text{greedy}}(Q^*)}(\overline{s}) = \mathbb{E}_{\overline{a} \sim \pi_{\text{greedy}}(Q^*)} Q^{\pi_{\text{greedy}}(Q^*)}(\overline{s}, \overline{a}) = \mathbb{E}_{\overline{a} \sim \pi_{\text{greedy}}(Q^*)} Q^*(\overline{s}, \overline{a})$$

By applying the definitions of $Q^*$ and $\pi_{\text{greedy}}$, we recover the definition of $V^*$. $\qquad \square$

Figure 4: Representation of $Q_\varepsilon^1$ (blue) and $Q_\varepsilon^2$ (yellow)

## A.4 Theorem 2

*Proof.* In the trivial case $|\mathcal{A}| = 1$, there exits only one policy $\pi$ and $\mathcal{T} = \mathcal{T}^\pi$, which is a contraction by Proposition 1.

In the general case $|\mathcal{A}| \geq 2$, we can build the following counter-example:

Let $(\mathcal{S}, \mathcal{A}, P, R_r, R_c)$ be a BMDP. For any $\varepsilon > 0$, we define $Q_\varepsilon^1$ and $Q_\varepsilon^2$ as:

$$Q_\varepsilon^1(\overline{s}, \overline{a}) = \begin{cases} (0,0), & \text{if } a = a_0 \\ \left(\frac{1}{\gamma}, \varepsilon\right), & \text{if } a \neq a_0 \end{cases}$$

$$Q_\varepsilon^2(\overline{s}, \overline{a}) = \begin{cases} (0,\varepsilon), & \text{if } a = a_0 \\ \left(\frac{1}{\gamma}, 2\varepsilon\right), & \text{if } a \neq a_0 \end{cases}$$

Then, $\|Q_1 - Q_2\|_\infty = \varepsilon$. $Q_\varepsilon^1$ and $Q_\varepsilon^2$ are represented in Figure 4.

But for $\overline{a} = (a, \beta_a)$ with $\beta_a = \varepsilon$, we have:

$$\|\mathcal{T}Q_\varepsilon^1(\overline{s}, \overline{a}) - \mathcal{T}Q_\varepsilon^2(\overline{s}, \overline{a})\|_\infty = \gamma \left\| \mathop{\mathbb{E}}_{\overline{s}' \sim \overline{P}(\overline{s}'|\overline{s}, \overline{a})} \mathop{\mathbb{E}}_{\overline{a}' \sim \pi_{\text{greedy}}(Q_\varepsilon^1)} Q_\varepsilon^1(\overline{s}', \overline{a}') - \mathop{\mathbb{E}}_{\overline{a}' \sim \pi_{\text{greedy}}(Q_\varepsilon^2)} Q_\varepsilon^2(\overline{s}', \overline{a}') \right\|_\infty$$

$$= \gamma \left\| \mathop{\mathbb{E}}_{\overline{s}' \sim \overline{P}(\overline{s}'|\overline{s}, \overline{a})} \left(\frac{1}{\gamma}, \varepsilon\right) - (0, \varepsilon) \right\|_\infty$$

$$= \gamma \frac{1}{\gamma} = 1$$

Hence,

$$\|\mathcal{T}Q_\varepsilon^1 - \mathcal{T}Q_\varepsilon^2\|_\infty \geq 1 = \frac{1}{\varepsilon}\|Q_1 - Q_2\|_\infty$$

In particular, there does not exist $L > 0$ such that:

$$\forall Q_1, Q_2 \in (\mathbb{R}^2)^{\overline{\mathcal{S}\mathcal{A}}}, \|\mathcal{T}Q^1 - \mathcal{T}Q^2\|_\infty \leq L\|Q^1 - Q^2\|_\infty$$

In other words, $\mathcal{T}$ is not a contraction for $\|\cdot\|_\infty$. $\qquad\square$

## A.5 Theorem 3

**Remark.** *This proof makes use of insights detailed in the proof of Proposition 3 (Appendix A.6), which we recommend the reader to consult first.*

*Proof.* We now study the contractivity of $\mathcal{T}$ when restricted to the functions of $\mathcal{L}_\gamma$ defined as follows:

$$\mathcal{L}_\gamma = \left\{ \begin{array}{l} Q \in (\mathbb{R}^2)^{\overline{\mathcal{SA}}} \text{ s.t. } \exists L < \frac{1}{\gamma} - 1 : \forall \overline{s} \in \overline{\mathcal{S}}, \overline{a}_1, \overline{a}_2 \in \overline{\mathcal{A}}, \\ |Q_r(\overline{s}, \overline{a}_1) - Q_r(\overline{s}, \overline{a}_2)| \le L|Q_c(\overline{s}, \overline{a}_1) - Q_c(\overline{s}, \overline{a}_2)| \end{array} \right\} \tag{19}$$

That is, for all state $\overline{s}$, the set $Q(\overline{s}, \overline{\mathcal{A}})$ plot in the $(Q_c, Q_r)$ plane must be the *graph* of a $L$-Lipschitz function, with $L < 1/\gamma - 1$.

We impose such structure for the following reason: the counter-example presented above prevented contraction because it was a pathological case in which the slope of $Q$ can be arbitrary large. As a consequence, when solving $Q_r^*$ such that $Q_c^* = \beta$, a vertical slice of a $\| \cdot \|_\infty$ ball around $Q_1$ (which must contain $Q_2$) can be arbitrary large as well.

This proof makes use of insights detailed in the proof of Proposition 3, which we recommend the reader to consult first.

We denote $\mathcal{B}(Q, R)$ the ball of centre $Q$ and radius $R$ for the $\| \cdot \|_\infty$-norm:

$$\mathcal{B}(Q, R) = \{Q' \in (R^2)^{\overline{\mathcal{SA}}} : \|Q - Q'\|_\infty \le R\}$$

We give the three main steps required to show that $\mathcal{T}$ restricted to $\mathcal{L}_\gamma$ is a contraction. Given $Q^1, Q^2 \in \mathcal{L}_\gamma$, show that:

1. $Q^2 \in \mathcal{B}(Q^1, R) \implies \mathcal{F}^2 \in \mathcal{B}(\mathcal{F}^1, R), \forall \overline{s} \in \overline{\mathcal{S}}$, where $\mathcal{F}$ is the top frontier of the convex hull of undominated points, as defined in Appendix A.6.

2. $Q \in \mathcal{L}_\gamma \implies \mathcal{F}$ is the graph of a $L$-Lipschitz function, $\forall \overline{s} \in \overline{\mathcal{S}}$.

3. taking the slice $Q_c = \beta$ of a ball $\mathcal{B}(\mathcal{F}, R)$ with $\mathcal{F}$ $L$-Lipschitz results in an interval on $Q_r$ of range at most $(L + 1)R$

These three steps will allow us to control $Q_r^{2*} - Q_r^{1*}$ as a function of $R = \|Q^2 - Q^1\|_\infty$.

**Step 1:** we want to show that if $Q^1$ and $Q^2$ are close, then $\mathcal{F}^1$ are $\mathcal{F}^2$ are close as well in the following sense:

$$\mathcal{F}^2 \in \mathcal{B}(\mathcal{F}^1, R) \iff d(\mathcal{F}^1, \mathcal{F}^2) \le R \iff \max_{q^2 \in \mathcal{F}^2} \min_{q^1 \in \mathcal{F}^1} \|q^2 - q^1\|_\infty \le R \tag{20}$$

Assume $Q^2 \in \mathcal{B}(Q^1, R)$, we show by contradiction that $\mathcal{F}^2 \in B(\mathcal{F}^1, R)$. Indeed, assume there exists $q^1 \in \mathcal{F}^1$ such that $\mathcal{F}^2 \cap B(q^1, R) = \emptyset$. Denote $q^2$ the unique point of $\mathcal{F}^2$ such that $q_c^2 = q_c^1$. By construction of $q^1$, we know that $\|q^1 - q^2\|_\infty > R$. There are two possible cases:

- $q_r^2 > q_r^1$: this also directly implies that $q_r^2 > q_r^1 + R$. But $q^2 \in \mathcal{F}^2$, so there exist $q_1^2, q_2^2 \in Q^2, \lambda \in \mathbb{R}$ such that $q^2 = (1 - \lambda)q_1^2 + \lambda q_2^2$. But since $Q^2 \in B(Q^1, R)$, there also exist $q_1^1, q_2^1 \in Q^1$ such that $\|q_1^1 - q_1^2\|_\infty \le R$ and $\|q_2^1 - q_2^2\|_\infty \le R$, and in particular $q_{1r}^1 \ge q_{1r}^2 - R$ and $q_{2r}^1 \ge q_{2r}^2 - R$. But then, the point $q^{1'} = (1 - \mu)q_1^1 + \mu q_2^1$ with $\mu = (q_c^2 - q_{1c}^1)/(q_{2c}^2 - q_{1c}^1)$ verifies $q_c^{1'} = q_c^1$ and $q_r^{1'} \ge q_r^2 - R > q_r^1$ which contradicts the definition of $q_1 \in \mathcal{F}^1$ as defined in (25).

- $q_r^2 < q_r^1$: then the same reasoning can be applied by simply swapping the indexes 1 and 2.

We have shown that $\mathcal{F}^2 \in B(\mathcal{F}^1, R)$. This is illustrated in Figure 5: given a function $Q^1$, we show the locus $\mathcal{B}(Q_1, R)$ of $Q^2$. We then draw $\mathcal{F}^1$ the top frontier of the convex hull of $Q^1$ and alongside the locus of all possible $\mathcal{F}^2$, which belong to a ball $\mathcal{B}(\mathcal{F}^1, R)$.

**Step 2:** We want to show that if $Q \in \mathcal{L}_\gamma$, $\mathcal{F}$ is the graph of an $L$-Lipschitz function:

$$\forall q^1, q^2 \in \mathcal{F}, |q_r^2 - q_r^1| \le |q_c^2 - q_c^1| \tag{21}$$

Figure 5: We represent the range of possible solutions $Q_r^{2,*}$ for any $Q^2 \in \mathcal{B}(Q^1)$, given $Q_1 \in \mathcal{L}_\lambda$

Let $Q \in \mathcal{L}_\gamma$ and $\overline{s} \in \overline{\mathcal{S}}$, $\mathcal{F}$ the corresponding top frontier of convex hull. For all $q^1, q^2 \in \mathcal{F}$, $\exists \lambda, \mu \in [0,1]$, $q^{11}, q^{12}, q^{21}, q^{22} \in Q(\overline{s}, \overline{\mathcal{A}})$ such that $q^1 = (1-\lambda)q^{11} + \lambda q^{12}$ and $q^2 = (1-\mu)q^{21} + \mu q^{22}$. Without loss of generality, we can assume $q_c^{11} \le q_c^{12}$ and $q_c^{21} \le q_c^{22}$. We also consider the worst case in terms of maximum $q_r$ deviation: $q_c^{12} \le q_c^{21}$. Then the maximum increment $q_r^2 - q_r^1$ is:

$$\|q_r^2 - q_r^1\| \le \|q_r^{12} - q_r^1\| + \|q_r^{21} - q_r^{12}\| + \|q_r^2 - q_r^{21}\|$$
$$= (1-\lambda)\|q_r^{12} - q_r^{11}\| + \|q_r^{21} - q_r^{12}\| + \mu\|q_r^{22} - q_r^{21}\|$$
$$\le (1-\lambda)L\|q_c^{12} - q_c^{11}\| + L\|q_c^{21} - q_c^{12}\| + \mu L\|q_c^{22} - q_c^{21}\|$$
$$= L\|q_c^{12} - q_c^1\| + L\|q_c^{21} - q_c^{12}\| + L\|q_c^2 - q_c^{21}\|$$
$$= L\|q_c^2 - q_c^1\|$$

This can also be seen in Figure 5: the maximum slope of the $\mathcal{F}^1$ is lower than the maximum slope between two points of $Q^1$.

**Step 3:** Let $\mathcal{F}_1$ be a L-Lipschitz set as defined in (21), and consider a ball $\mathcal{B}(\mathcal{F}_1, R)$ around it as defined in (20).

We want to bound the optimal reward value $Q_r^{2*}$ under constraint $Q_c^{2*} = \beta$ (regular case in Appendix A.6 where the constraint is saturated), for any $\mathcal{F}^2 \in \mathcal{B}(\mathcal{F}_1, R)$. This quantity is represented as a red double-ended arrow in Figure 5.

Because we are only interested in what happens locally at $Q_c = \beta$, we can zoom in on Figure 5 and only consider a thin $\varepsilon$-section around $\beta$. In the limit $\varepsilon \to 0$, this section becomes the tangent to $\mathcal{F}^1$ at $Q_c^1 = \beta$. It is represented in Figure 6, from which we derive a geometrical proof:

$$\Delta Q_r^{2*} = b + c$$
$$\le La + c \qquad\qquad (\mathcal{F}^1\ L\text{-Lipschitz})$$
$$= 2LR + 2R = 2R(L+1)$$

Hence,

$$|Q_r^{2*} - Q_r^{1*}| \le \frac{\Delta Q_r^{2*}}{2} = R(L+1)$$

and $Q_c^{1*} = Q_c^{2*} = \beta$. Consequently, $\|Q^{2*} - Q^{1*}\|_\infty \le (L+1)R$

Finally, consider the edge case in Appendix A.6: the constraint is not active, and the optimal value is simply $\arg\max_{q \in \mathcal{F}} q^r$. In particular, since we showed that $\mathcal{F}^2 \in B(\mathcal{F}^1, R)$, and since $Q^{2*} \in \mathcal{F}^2$, there exist $q^1 \in \mathcal{F}^1 : \|Q^{2*} - q^1\|_\infty \le R$ and in particular $Q_r^{1*} \ge q_r^1 \ge Q_r^{2*} - R$. Reciprocally, by the same reasoning, $Q_r^{2*} \ge Q_r^{1*} - R$. Hence, we have that $|Q_r^{2*} - Q_r^{1*}| \le R \le R(L+1)$.

**Wrapping it up:**

Figure 6: We represent a section $[\beta - \varepsilon, \beta + \varepsilon]$ of $\mathcal{F}^1$ and $B(\mathcal{F}^1, R)$. We want to bound the range of $Q_r^{2*}$.

We've shown that for any $Q^1, Q^2 \in \mathcal{L}_\gamma$, and all $\bar{s} \in \overline{\mathcal{S}}$, $\mathcal{F}^2 \in \mathcal{B}(\mathcal{F}^1, \|Q^2 - Q^1\|_\infty)$ and $\mathcal{F}^1$ is the graph of a $L$-Lipschitz function with $L < 1/\gamma - 1$. Moreover, the solutions of $\pi_{\text{greedy}}(Q^1)$ and $\pi_{\text{greedy}}(Q^2)$ at $\bar{s}$ are such that $\|Q^{2*} - Q^{1*}\|_\infty \leq (L+1)\|Q^2 - Q^1\|_\infty$.

Hence, for all $\bar{a}$,

$$\|\mathcal{T}Q^1(\bar{s}, \bar{a}) - \mathcal{T}Q^2(\bar{s}, \bar{a})\|_\infty = \gamma \left\| \mathop{\mathbb{E}}_{\bar{s}' \sim \overline{P}(\bar{s}'|\bar{s}, \bar{a})} \mathop{\mathbb{E}}_{\bar{a}' \sim \pi_{\text{greedy}}(Q^1)} Q^1(\bar{s}', \bar{a}') - \mathop{\mathbb{E}}_{\bar{a}' \sim \pi_{\text{greedy}}(Q^2)} Q^2(\bar{s}', \bar{a}') \right\|_\infty$$
$$= \gamma \left\| Q^{2*} - Q^{1*} \right\|_\infty$$
$$\leq \gamma(L+1)\|Q^2 - Q^1\|_\infty$$

Taking the sup on $\overline{\mathcal{S}\mathcal{A}}$,

$$\|\mathcal{T}Q^1 - \mathcal{T}Q^2\|_\infty \leq \gamma(L+1)\|Q^1 - Q^2\|_\infty$$

with $\gamma(L+1) < 1$. As a conclusion, $\mathcal{T}$ is a $\gamma(L+1)$-contraction on $\mathcal{L}_\gamma$. $\qquad\square$

### A.6 Proposition 3

**Definition 2.** *Let $A$ be a set, and $f$ a function defined on $A$. We define:*

- *Convex hull of $A$: $\mathcal{C}(A) = \{\sum_{i=1}^p \lambda_i a_i : a_i \in A, \lambda_i \in \mathbb{R}^+, \sum_{i=1}^p \lambda_i = 1, p \in \mathbb{N}\}$*

- *Convex edges of $A$: $\mathcal{C}^2(A) = \{\lambda a_1 + (1-\lambda)a_2 : a_1, a_2 \in A, \lambda \in [0,1]\}$*

- *Dirac distributions of $A$: $\delta(A) = \{\delta(a - a_0) : a_0 \in A\}$*

- *Image of $A$ by $f$: $f(A) = \{f(a) : a \in A\}$*

*Proof.* Let $\bar{s} = (s, \beta) \in \overline{\mathcal{S}}$ and $Q \in (\mathbb{R}^2)^{\overline{\mathcal{S}\mathcal{A}}}$. We recall the definition of $\pi_{\text{greedy}}$:

$$\pi_{\text{greedy}}(\bar{a}|\bar{s}; Q) \in \arg\min_{\rho \in \Pi_r^Q} \mathop{\mathbb{E}}_{\bar{a} \sim \rho} Q_c(\bar{s}, \bar{a}) \tag{12a}$$

$$\text{where} \quad \Pi_r^Q = \arg\max_{\rho \in \mathcal{M}(\overline{A})} \mathop{\mathbb{E}}_{\bar{a} \sim \rho} Q_r(\bar{s}, \bar{a}) \tag{12b}$$

$$\text{s.t.} \quad \mathop{\mathbb{E}}_{\bar{a} \sim \rho} Q_c(\bar{s}, \bar{a}) \leq \beta \tag{12c}$$

Note that any policy in the $\arg\min$ in (12a) is suitable to compute $\mathcal{T}$. We first reduce the set of candidate optimal policies. Consider the problem described in (12b),(12c): it can be seen as a

single-step CMDP problem with reward $R_r = Q_r$ and cost $R_c = Q_c$. By (Theorem 4.4 Beutler and Ross, 1985), we know that the solutions are mixtures of two deterministic policies. Hence, we can replace $\mathcal{M}(\mathcal{A})$ by $\mathcal{C}^2(\delta(\overline{\mathcal{A}}))$ in (12b).

Moreover, remark that:

$$\{\underset{\overline{a}\sim\rho}{\mathbb{E}}\ Q(\overline{s},\overline{a}) : \rho \in \mathcal{C}^2(\delta(\overline{\mathcal{A}}))\} = \{\underset{\overline{a}\sim\rho}{\mathbb{E}}\ Q(\overline{s},\overline{a}) : \rho = (1-\lambda)\delta(\overline{a}-\overline{a}_1) + \lambda\delta(\overline{a}-\overline{a}_2), \overline{a}_1, \overline{a}_2 \in \overline{\mathcal{A}}, \lambda \in [0,1]\}$$
$$= \{(1-\lambda)Q(\overline{s},\overline{a}_1) + \lambda Q(\overline{s},\overline{a}_2), \overline{a}_1, \overline{a}_2 \in \overline{\mathcal{A}}, \lambda \in [0,1]\}$$
$$= \mathcal{C}^2(Q(\overline{s},\overline{\mathcal{A}}))\}$$

Hence, the problem (12b), (12c) has become:

$$\widetilde{\Pi}_r^Q = \underset{(q_r,q_c)\in\mathcal{C}^2(Q(\overline{s},\overline{\mathcal{A}}))}{\arg\max}\ q_r \quad \text{s.t.} \quad q_c \leq \beta$$

and the solution of $\pi_{\text{greedy}}$ is $q^* = \arg\min_{q\in\widetilde{\Pi}_r^Q} q_c$.

The original problem in the space of actions $\overline{\mathcal{A}}$ is now expressed in the space of values $Q(\overline{s}, \overline{\mathcal{A}})$ (which is why we use $=$ instead of $\in$ before $\arg\min$ here).

We further restrict the search space of $q^*$ following two observations:

1. $q^*$ belongs to the *undominated* points $\mathcal{C}^2(Q^-)$:

$$Q^+ = \{(q_c, q_r) : q_c > q_c^{\pm} = \min_{q^+} q_c^+ \text{ s.t. } q^+ \in \arg\max_{q\in Q(\overline{s},\overline{\mathcal{A}})} q_r\} \qquad (23)$$

$$Q^- = Q(\overline{s}, \overline{\mathcal{A}}) \setminus Q^+ \qquad (24)$$

Denote $q^* = (1-\lambda)q^1 + \lambda q^2$, with $q^1, q^2 \in Q(\overline{s}, \overline{\mathcal{A}})$. There are three possible cases:

   (a) $q^1, q^2 \notin Q^-$. Then $q_c^* = (1-\lambda)q_c^1 + \lambda q_c^2 > q_c^{\pm}$. But then $q_c^{\pm} < q_c^* \leq \beta$ so $q^{\pm} \in \widetilde{\Pi}_r^Q$ with a strictly lower $q_c$ than $q^*$, which contradicts the $\arg\min$.
   (b) $q^1 \in Q^-, q^2 \notin Q^-$. But then consider the mixture $q^{\top} = (1-\lambda)q^1 + \lambda q^{\pm}$. Since $q_r^{\pm} \geq q_r^2$ and $q_r^{\pm} < q_c^2$, we also have $q_r^{\top} \geq q_r^*$ and $q_c^{\top} < q_c^*$, which also contradicts the $\arg\min$.
   (c) $q^1, q^2 \in Q^-$ is the only remaining possibility.

2. $q^*$ belongs to the *top frontier* $\mathcal{F}$:

$$\mathcal{F}_Q = \{q \in \mathcal{C}^2(Q^-) : \nexists q' \in \mathcal{C}^2(Q^-) : q_c = q_c' \text{ and } q_r < q_r'\} \qquad (25)$$

Trivially, otherwise q' would be a better candidate than $q^*$.

Let us characterise this frontier $\mathcal{F}$. It is both:

1. the *graph of a non-decreasing function*: $\forall q^1, q^2 \in \mathcal{F}$ such that $q_c^1 \leq q_c^2$ then $q_r^1 \leq q_r^2$.
   By contradiction, if we had $q_r^1 > q_r^2$, we could define $q^{\top} = (1-\lambda)q^1 + \lambda q^{\pm}$ where $q^{\pm}$ is the dominant point as defined in (23). By choosing $\lambda = (q_c^2 - q_c^1)/(q_c^{\pm} - q_c^1)$ such that $q_c^{\top} = q_c^2$, then since $q_r^{\pm} \geq q_r^1 > q_r 2$ we also have $q_r^{\top} > q_r^2$ which contradicts $q^2 \in \mathcal{F}$.

2. the *graph of a concave function*: $\forall q^1, q^2, q^3 \in \mathcal{F}$ such that $q_c^1 \leq q_c^2 \leq q_c^3$ with $\lambda$ such that $q_c^2 = (1-\lambda)q_c^1 + \lambda q_c^3$, then $q_r^2 \geq (1-\lambda)q_r^1 + \lambda q_r^3$.
   Trivially, otherwise the point $q^{\top} = (1-\lambda)q^1 + \lambda q^3$ would verify $q_c^{\top} = q_c^2$ and $q_r^{\top} > q_r^2$, which would contradict $q^2 \in \mathcal{F}$.

We denote $\mathcal{F}_Q = \mathcal{F} \cap Q$. Clearly, $q^* \in \mathcal{C}^2(\mathcal{F}_Q)$: let $q^1, q^2 \in Q^-$ such that $q^* = (1-\lambda)q^1 + \lambda q^2$. First, $q^1, q^2 \in Q^- \subset \mathcal{C}^2(Q^-)$. Then, by contradiction, if there existed $q^{1'}$ or $q^{2'}$ with equal $q_c$ and strictly higher $q_r$, again we could build an admissible mixture $q^{\top} = (1-\lambda)q^{1'} + \lambda q^{2'}$ strictly better than $q^*$.

$q^*$ can be written as $q^* = (1-\lambda)q^1 + \lambda q^2$ with $q^1, q^2 \in \mathcal{F}_Q$ and, without loss of generality, $q_c^1 \leq q_c^2$.

Figure 7: Neural Network for $Q$-functions approximation when $\mathcal{S} = \mathbb{R}^2$ and $|\mathcal{A}| = 2$.

**Regular case:** there exists $q^0 \in \mathcal{F}_Q$ such that $q_c^0 \geq \beta$.

Then $q^1$ and $q^2$ must flank the budget: $q_c^1 \leq \beta \leq q_c^2$. Indeed, by contradiction, if $q_c^2 \geq q_c^1 > \beta$ then $q_c^* > \beta$ which contradicts $\Pi_r^Q$. Conversely, if $q_c^1 \leq q_c^2 < \beta$ then $q^* < \beta \leq q_c^0$, which would make $q^*$ a worse candidate than $q^\top = (1 - \lambda)q^* + \lambda q^0$ when $\lambda$ is chosen such that $q_c^\top = \beta$, and contradict $\Pi_r^Q$ again.

Because $\mathcal{F}$ is the graph of a non-decreasing function, $\lambda$ should be as high as possible, as long as the budget $q^* \leq \beta$ is respected. We reach the highest $q_r^*$ when $q_c^* = \beta$, that is: $\lambda = (\beta - q_c^1)/(q_c^2 - q_c^1)$.

It remains to show that $q^1$ and $q^2$ are two successive points in $\mathcal{F}_Q$: $\nexists q \in \mathcal{F}_Q \backslash \{q^1, q^2\} : q_c^1 \leq q_c \leq q_c^2$. Otherwise, as $\mathcal{F}$ is the graph of a concave function, we would have $q_r \geq (1 - \mu)q_r^1 + \mu q_r^2$. $q_r$ cannot be strictly greater than $(1 - \mu)q_r^1 + \mu q_r^2$ which would contradict $q^*$, but it can still be equal, which means the tree points $q, q^1, q^2$ are aligned. In fact, every points aligned with $q^1$ and $q^2$ can also be used to construct mixtures resulting in $q^*$, but among these solutions we can still choose $q^1$ and $q^2$ as the two points in $\mathcal{F}_Q$ closest to $q^*$.

**Edge case:** $\forall q \in \mathcal{F}_Q, q_c < \beta$. Then $q^* = \arg\max_{q \in \mathcal{F}} q_r = q^{\pm} = \arg\max_{q \in Q^-} q_r$

□

# B   Scalable Implementation of BFTQ

We recall the scalable version of BFTQ in Algorithm 5 and the architecture of the neural network Figure 7.

**Remark 2.** *Because of the budget dynamics $\beta' = \beta_a$ in (2), we can see in (5) that for all $\overline{s} = (s, \beta)$ and $\overline{a} = (a, \beta_a)$, $Q^\pi(\overline{s}, \overline{a})$ only depends on $s, a, \beta_a$ and not on $\beta$. We will slightly abuse notations and sometimes denote in the following $Q^\pi(s, a, \beta_a) \overset{\text{def}}{=} Q^\pi(\overline{s}, \overline{a})$.*

# C   The Lagrangian Relaxation Baseline

As explained on Figure 8, the optimal deterministic policy can be obtained by a line-search on the Lagrange multiplier values $\lambda$.

Then, according to Beutler and Ross (1985, Theorem 4.4), the optimal policy is a randomised mixture of two deterministic policies: the safest deterministic policy that violates the constraint $\pi_{\lambda_-}$ and the riskier of the feasible ones $\pi_{\lambda_+}$.

Fitted-Q (FTQ) (Ernst *et al.*, 2005; Riedmiller, 2005) can be easily adapted for continuous states CMDP and BMDP through this methodology, but given the high variance it requires a lot of sim-

**Algorithm 5:** Scalable BFTQ

**Data:** $\mathcal{D}$, $\widetilde{\mathcal{B}}$ a finite subset of $\mathcal{B}$, $\gamma$, a model $Q \in (\mathbb{R}^2)^{S\mathcal{A}}$, a regression algorithm $\texttt{fit}$, a set of CPU workers $W$

**Result:** $Q^*$

1   $Q \leftarrow 0$
2   $X \leftarrow \{s_i, a_i, \beta_{a_i}\}_{i \in [0, |\mathcal{D}|]}$
3   $S' \leftarrow \{s_i'\}_{i \in [0, |\mathcal{D}|]}$
4   **repeat**
5      Evaluate $Q(S', \mathcal{A}, \widetilde{\mathcal{B}})$ in a single forward pass
6      Split $\mathcal{D}$ among workers: $\mathcal{D} = \cup_{w \in W} \mathcal{D}_w$
7      **for** $w \in W$ **do**                           `// Run in parallel`
8          **for** $(\cdot, \cdot, \beta_{a_i}, R_{r_i}, R_{c_i}, s_i') \in \mathcal{D}$ **do**
9              $\mathcal{P} \leftarrow \{(Q_c(s_i', \mathcal{A}, \widetilde{\mathcal{B}}), Q_r(s_i', \mathcal{A}, \widetilde{\mathcal{B}}))\}$
10             $\mathcal{P}$.prune()        `// Remove all dominated points`
11             $\mathcal{H} \leftarrow \texttt{convex\_hull}(\mathcal{P}).\text{vertices}()$       `// in cw order`
12             $k \leftarrow \min\{k : \beta_i \geq q_c \text{ with } (q_c, q_r) = \mathcal{H}[k]\}$
13             $q_c^2, q_r^2, q_c^1, q_r^1 \leftarrow \mathcal{H}[k], \mathcal{H}[k-1]$
14             $p \leftarrow (\beta_{a_i} - q_a^1)/(q_c^2 - q_c^1)$
15             $Y_c^{w,i} \leftarrow R_{c_i} + \gamma((1-p)q_c^1 + pq_c^2)$
16             $Y_r^{w,i} \leftarrow R_{r_i} + \gamma((1-p)q_r^1 + pq_r^2)$
17          **end**
18      **end**
19      Join the results: $Y \leftarrow \cup_{w \in W}(Y_c^w, Y_r^w)$
20      $Q \leftarrow \texttt{fit}(X, Y)$
21   **until** *convergence*

Figure 8: Calibration of a penalty multiplier according to the budget $\beta$. The optimal multiplier $\lambda_{\text{avg}}^*$ is the smallest one to satisfy the budget constraint on average. Safer policies can also be selected according to the largest deviation from this mean cost.

ulations to get a proper estimate of the calibration curve. Our purpose is to avoid this calibration phase.

## D   Experiments

### D.1   Examples of different exploration strategies

We compare two approaches for constructing a batch of samples. The animations at this https URL display the trajectories collected in each intermediate sub-batch. The first row corresponds to a classical risk neutral epsilon-greedy exploration policy while the second row showcases a risk-

sensitive exploration strategy introduced in the paper. Each animation corresponds to a different seed.

## D.2 Examples of BFTQ policies executions

We display the evolution in the budgeted policy behaviour with respect to the budget on different environments. The policies have been learnt with a risk-sensitive exploration.

**Highway-Env** On the `highway-env` , the budgeted agents display a wide variety of behaviours. Animations are displayed at this https URL. When $\beta = 1$, the ego-vehicle drives in a very aggressive style: it immediately switches to the opposite lane and drives as fast as possible to pass slower vehicles, swiftly changing lanes to avoid incoming traffic. On the contrary when $\beta = 0$, the ego-vehicle is conservative: it stays on its lane and drives at a low velocity. With intermediate budgets such as $\beta = 0.2$, the agent sometimes decides to overtake its front vehicle but promptly steers back to its original lane afterwards.

**Slot-filling**

**Remark on the `slot-filling` environment** When receiving an utterance, the system can either understand it ($\mu = \mu_u$) or misunderstand it ($\mu = \mu_m$) with a fixed probability called the sentence error rate $ser$. Then, the speech recognition score is simulated (Khouzaimi *et al.*, 2015): $srs = (1 + \exp(-x))^{-1}$ with $x \sim N(\mu, \sigma)$. It's the confidence score of the natural language understanding module about the last utterance. Note that here are no recognition errors ($ser = 0$ and $srs = 1$) when the user provides information using the numeric pad.

In Table 1, we display two dialogues done with the same BFTQ policy on `slot-filling`. The policy is given two budgets to respect in expectation, $\beta = 0$ and $\beta = 0.5$. For $\beta = 0$, one can see that the system never uses the `ask_num_pad` action. Instead, it uses `ask_oral` , an action subject to recognition errors. The system keeps asking for the same slot 2, because it has the lowest speech recognition score. It eventually summarises the form to the user, but then reaches the maximum dialogue length and thus faces a dialogue failure. For $\beta = 0.5$, the system first asks in a safe way, with `ask_oral`. It may want to `ask_num_pad` if one of the speech recognition score is low. Then, the system proceeds to a confirmation of the slot values. If it is incorrect, the system continues the dialogue using unsafe the `ask_num_pad` action to be certain of the slot values.

**Corridors** Animations are displayed at this https URL for the `corridors` environment. When the budget is low, the agent takes the safest path on the left. When the budget increases, it gradually switches to the other lane, earning higher rewards but also costs. This gradual process could not be achieved with a deterministic policy as it would chose either one path or the other. Each animation corresponds to a different seed.

## D.3 Reproducibility

The following section displays environments and algorithms parameters and instructions to reproduce the exact same results displayed in Section 5.

### D.3.1 Algorithm parameters

All algorithm parameters are displayed in Table 2,Table 3 and Table 4.

**A note on the parameters search** We performed a shallow grid-search for the classic Neural-Network parameters. Most of the parameters don't have a strong influence on the results, however in the `slot-filling` environment, the choice of the regulation weight is decisive.

### D.3.2 Environments Parameters

All environments parameters are displayed in Table 5, Table 6 and Table 7.

| turn | $\beta = 0$ | $\beta = 0.5$ |
|---|---|---|
| turn 0 | valid slots : [0, 0, 0]<br>srs : [ None None None ]<br>system says ASK_ORAL(1)<br>user says INFORM | valid slots : [0, 0, 0]<br>srs : [ None None None ]<br>system says ASK_ORAL(2)<br>user says INFORM |
| turn 1 | valid slots : [0, 0, 0]<br>srs : [ None 0.48 None ]<br>system says ASK_ORAL(2)<br>user says INFORM | valid slots : [0, 0, 1]<br>srs : [ None None 0.56 ]<br>system says ASK_ORAL(0)<br>user says INFORM |
| turn 2 | valid slots : [0, 0, 0]<br>srs : [ None 0.48 0.22 ]<br>system says ASK_ORAL(0)<br>user says INFORM | valid slots : [0, 0, 1]<br>srs : [ 0.30 None 0.56 ]<br>system says ASK_ORAL(1)<br>user says INFORM |
| turn 3 | valid slots : [0, 0, 0]<br>srs : [ 0.62 0.48 0.22 ]<br>system says ASK_ORAL(2)<br>user says INFORM | valid slots : [0, 0, 1]<br>srs : [ 0.30 0.54 0.56 ]<br>system says ASK_ORAL(0)<br>user says INFORM |
| turn 4 | valid slots : [0, 0, 0]<br>srs : [ 0.62 0.48 0.66 ]<br>system says ASK_ORAL(1)<br>user says INFORM | valid slots : [0, 0, 1]<br>srs : [ 0.68 0.54 0.56 ]<br>system says ASK_NUM_PAD(1)<br>user says INFORM |
| turn 5 | valid slots : [0, 1, 0]<br>srs : [ 0.62 0.56 0.66 ]<br>system says ASK_ORAL(2)<br>user says INFORM | valid slots : [0, 1, 1]<br>srs : [ 0.68 1.00 0.56 ]<br>system says SUMMARIZE_AND_INFORM<br>user says DENY_SUMMARIZE |
| turn 6 | valid slots : [0, 1, 0]<br>srs : [ 0.62 0.56 0.14 ]<br>system says ASK_ORAL(2)<br>user says INFORM | valid slots : [0, 1, 1]<br>srs : [ 0.68 1.00 0.56 ]<br>system says ASK_NUM_PAD(2)<br>user says INFORM |
| turn 7 | valid slots : [0, 1, 1]<br>srs : [ 0.62 0.56 0.30 ]<br>system says ASK_ORAL(2)<br>user says INFORM | valid slots : [0, 1, 1]<br>srs : [ 0.68 1.00 1.00 ]<br>system says SUMMARIZE_AND_INFORM<br>user says DENY_SUMMARIZE |
| turn 8 | valid slots : [0, 1, 1]<br>srs : [ 0.62 0.56 0.49 ]<br>system says ASK_ORAL(2)<br>user says INFORM | valid slots : [0, 1, 1]<br>srs : [ 0.68 1.00 1.00 ]<br>system says ASK_NUM_PAD(0)<br>user hangs up ! |
| turn 9 | valid slots : [0, 1, 1]<br>srs : [ 0.62 0.56 0.65 ]<br>system says SUMMARIZE_AND_INFORM<br>max size reached ! | |

Table 1: Two dialogues generated by a safe policy ($\beta = 0$) on the left and a risky one ($\beta = 0.5$) on the right.

| Parameters | BFTQ(risk-sensitive) | BFTQ(risk-neutral) |
| --- | --- | --- |
| architecture | 256x128x64 | 256x128x64 |
| regularisation | 0.001 | 0.001 |
| activation | relu | relu |
| size beta encoder | 3 | 3 |
| initialisation | xavier | xavier |
| loss function | L2 | L2 |
| optimizer | adam | adam |
| learning rate | 0.001 | 0.001 |
| epoch (NN) | 1000 | 5000 |
| normalize reward | true | true |
| epoch (FTQ) | 12 | 12 |
| $\widetilde{\mathcal{B}}$ | 0:0.01:1 | - |
| $\gamma$ | 1 | 1 |
| $N = |\mathcal{D}|$ | 5000 | 5000 |
| $N_{\text{minibatch}}$ | 10 | 10 |
| $N_{\text{seeds}}$ | 4 | 4 |
| $N_{\text{test}}$ | 1000 | 1000 |
| decay epsilon scheduling | 0.001 | 0.001 |

Table 2: Algorithms parameters for `Corridors`

| Parameters | BFTQ | FTQ |
| --- | --- | --- |
| architecture | 256x128x64 | 128x64x32 |
| regularisation | 0.0005 | 0.0005 |
| activation | relu | relu |
| size beta encoder | 50 | - |
| initialisation | xavier | xavier |
| loss function | L2 | L2 |
| optimizer | adam | adam |
| learning rate | 0.001 | 0.001 |
| epoch (NN) | 5000 | 5000 |
| normalize reward | true | true |
| epoch (FTQ) | 11 | 11 |
| $\widetilde{\mathcal{B}}$ | 0:0.01:1 | - |
| $\gamma$ | 1 | 1 |
| $N = |\mathcal{D}|$ | 5000 | 5000 |
| $N_{\text{minibatch}}$ | 10 | 10 |
| $N_{\text{seeds}}$ | 6 | 6 |
| $N_{\text{test}}$ | 1000 | 1000 |
| decay epsilon scheduling | 0.001 | 0.001 |

Table 3: Algorithms parameters for `Slot-Filling`

| Parameters | BFTQ | FTQ |
|---|---|---|
| architecture | 256x128x64 | 128x64x32 |
| regularisation | 0.0005 | 0 |
| activation | relu | relu |
| size beta encoder | 50 | - |
| initialisation | xavier | xavier |
| loss function | L2 | L2 |
| optimizer | adam | adam |
| learning rate | 0.001 | 0.01 |
| epoch (NN) | 5000 | 400 |
| normalize reward | true | true |
| epoch (FTQ) | 15 | 15 |
| $\widetilde{\mathcal{B}}$ | 0:0.01:1 | - |
| $\gamma$ | 0.9 | 0.9 |
| $N = |\mathcal{D}|$ | 10000 | 10000 |
| $N_{\text{minibatch}}$ | 10 | 10 |
| $N_{\text{seeds}}$ | 10 | 10 |
| $N_{\text{test}}$ | 150 | 150 |
| decay epsilon scheduling | 0.0003 | 0.0003 |

Table 4: Algorithms parameters for `Highway-Env`

**State-Space**   The states $s$ (from $\overline{s} = (s, \beta)$) of the agent are described in the following:

- `Corridors`: $s = (x, y)$ where $x$ and $y$ are the 2D coordinates of the agent.
- `Slot-Filling`: $s = (\text{srs}, \min, a_u, a_s, t)$ where srs is a vector of the speech recognition score for each slot, min is a one hot vector describing the minimum of the srs vector, $a_u$ is a one hot vector of the last user dialogue act and $a_s$ is the one hot vector of the last system dialogue act. Finally $t \in [0, 1]$ is the fraction of the current turn with the maximum number of turns authorised.
- `Highway-Env`: the positions $(x, y)$ and velocities $(\dot{x}, \dot{y})$ of every vehicle on the road.

| Parameter | Description | Value |
|---|---|---|
| - | Size of the environment | 7 x 6 |
| - | Standard deviation of the Gaussian noise applied to actions | (0.25,0.25) |
| H | Episode duration | 9 |

Table 5: Parameters of `Corridors`

| Parameter | Description | Value |
|---|---|---|
| ser | Sentence Error Rate | 0.6 |
| $\mu_m$ | Gaussian mean for misunderstanding | -0.25 |
| $\mu_u$ | Gaussian mean for understanding | 0.25 |
| $\sigma$ | Gaussian standard deviation | 0.6 |
| $p$ | Probability of hang-up | 0.25 |
| H | Episode duration | 10 |
| - | Number of slots | 3 |

Table 6: Parameters of `Slot-Filling`

### D.3.3   Instructions for reproducibility

To reproduce the result displayed in Section 5, first install the following conventional libraries for python3: pycairo, numpy, scipy and pytorch. Then, execute the commands in Figure 9 on a Linux

| Parameter | Description | Value |
|-----------|-------------|-------|
| $N_v$ | Number of vehicles | 2 - 6 |
| $\sigma_p$ | Standard deviation of vehicles initial positions | 100 m |
| $\sigma_v$ | Standard deviation of vehicles initial velocities | 3 m/s |
| H | Episode duration | 15 s |

Table 7: Parameters of `highway-env`

Listing 1: bash version

```bash
# Install highway-env
pip3 install --user git+https://github.com/eleurent/rl-agents
# clone the phd_code repository
git clone https://github.com/ncarrara/budgeted-rl
# Change python path to the path of this repository
export PYTHONPATH="budgeted-rl/ncarrara"
# Navigate to budgeted-rl folder
cd budgeted-rl/ncarrara/budgeted-rl
# Run main script using any config file
# Choose the range of seeds you want to test on
python3 main/egreedy/main-egreedy.py config/slot-filling.json 0 6
python3 main/egreedy/main-egreedy.py config/corridors.json 0 4
python3 main/egreedy/main-egreedy.py config/highway-two-way.json 0 10
```

Figure 9: Instructions to reproduce experiments

Operating System. The Graphic Processing Unit used for experiments is an NVIDIA `GeForce GTX 1080 Ti` and the Computational Processing Unit is an Intel `Xeon E7`. Then, follow the instructions in Figure 9. For more details, check this https URL.

## E The machine learning reproducibility checklist

For all models and algorithms presented, indicate if you include:

- A clear description of the mathematical setting, algorithm, and/or model:
  - **yes**, see Section 1, Section 2, Section 3, Section 3.2 and Appendix B.
- An analysis of the complexity (time, space, sample size) of any algorithm:
  - **yes**, see Section 4.3.
- A link to a downloadable source code, with specification of all dependencies, including external libraries:
  - **yes**, see Appendix D.3.3. All resources are available at this https URL.

For any theoretical claim, indicate if you include:

- A statement of the result:
  - **yes**, see Section 2 and Section 3.
- A clear explanation of any assumptions:
  - we make one assumption in Section 2. We assume the program is feasible for any state. If not, no algorithm would be able to solve it anyway.
- A complete proof of the claim:
  - **yes**, see Appendix A.

For all figures and tables that present empirical results, indicate if you include:

- A complete description of the data collection process, including sample size:
    - **yes**, see Section 5 and Appendix D.3.1.
- A link to a downloadable version of the dataset or simulation environment:
    - **yes**, all environments are fetch from public repositories, see Appendix D.3.3 for details.
- An explanation of any data that were excluded, description of any pre-processing step:
    - it's **not applicable** as data comes from simulated environments, so pre-processing steps are not needed.
- An explanation of how samples were allocated for training / validation / testing:
    - it's **not applicable**. The complete dataset is used for training. There is no need for validation set. Testing is performed in the true environment as in classical online learning approaches.
- The range of hyper-parameters considered, method to select the best hyper-parameter configuration, and specification of all hyper-parameters used to generate results:
    - **yes**, see Appendix D.3.1.
- The exact number of evaluation runs:
    - **yes**, see $N_{seeds}$ in the tables from Appendix D.3.1.
- A description of how experiments were run:
    - **yes**, see the two first paragraphs of Section 5.
- A clear definition of the specific measure or statistics used to report results:
    - **yes**, see Section 5.2.
- Clearly defined error bars:
    - **yes**, we plot 95% confidence intervals in all figures, see Section 5.2.
- A description of results with central tendency (e.g. mean) variation (e.g. stddev):
    - **yes**, we even observe less variability with our novel approach, see Section 5.2.
- A description of the computing infrastructure used:
    - The Graphic Processing Unit used for experiments is an NVIDIA `GeForce GTX 1080 Ti` and the Central Processing Unit is an Intel `Xeon E7`.