[Reviews · NeurIPS 2019]

Reviewer 1



The paper formulates a budgeted Markov decision process (BMDP) able to deal with large search spaces. The problem is framed as a multi-objective MDP considering the rewards and cost functions. This creates value functions for rewards and costs, and the framework searches for a minimum cost policy, within the maximum reward policies that satisfies the budget restrictions. The paper experimentally shows the performance of the proposed approach in three domains: corridor, spoken dialogue, and autonomous driving. The framework searches for a minimum cost policy given a maximum reward policy satisfying budget restrictions. In principle it could be framed the other way around searching for a maximum reward policy given minimum cost. Add a note of why this is not a good idea. When creating batch samples what happens when the budget is zero? The episodes ends with a negative reward? The proposed framework is not a contraction so in principle is not guaranteed to converge. The authors provide some possible explanation of why this was not an issue in the selected domains. In which cases should we not use the proposed approach? The convex hull policy algorithm is not very clearly described. How can the budget lie between two Q_c, is it not the case that all of them should respect the budget constraint? Typos: - provided Appendix A

Reviewer 2



SUMMARY: The authors considered budgeted Markov decision processes (BMDPs), which contain a notion of risk and a threshold to control that risk. Here, the authors seek to extend methods for BMDPs such that they can be applied to problems with continuous state spaces and unknown dynamics. In particular, they propose new extensions of deep reinforcement learning (DRL) methods such that they can be applied to BMDPs. The authors validate their approach on two simulated applications. STRENGTHS: * The paper is well motivated in that the authors seek to extend BMDPs to for continuous-state, infinite-horizon tasks, which could be an important step in trying to address some existing problems with "modern" RL approaches. * The authors make BMDPs in continuous-state, infinite-horizon tasks possible by providing a new algorithm, BFTQ, that enables learning in such cases. * The parts of the algorithm I understood seemed intuitive, and I found the figures and supplemental visualizations of experiments intuitive and compelling as well. WEAKNESSES: (A) To my read, the discussion of the algorithm and implementation are not quite complete in the sense that I still have several important questions that I did not feel were answered: (1) I am confused as to why/how the \beta evolves during the learning process and how this effects the goals one has when using a BMDP. Do we not care about how much risk is induced (or negative effects experienced) during training? Moreover, if the policy selects a new \beta at each new time step, how do we enforce a *specific* \beta when we wish to *use* the policy? (2) The convex hull procedure seems critical to being able to actually use the algorithm, but the explanation is lacking any intuitive interpretation. For example, it's not clear to me from 4.1 and Algorithm 3 exactly how "Budget \beta [is] always respected" (comment on Algorithm 3, line 9). Can the authors provide more explanation/intuition about what is going on here? (3) The authors state that "The pseudo-code of our exploration procedure is shown in Algorithm 4 in Appendix B." Since this component of your algorithm is one of the main hypotheses that is being validated, it should appear in the main paper. Suggest swapping for Algorithms 1/2 since these are basic extensions of existing techniques. (B) There is a mismatch between the introduction of the paper, which speaks generally to using BMDPs, and the actual experiments run, which show the benefit of using BMDPs *compared to* computing a set of solutions using existing techniques like FTQ(\lambda). The introduction needs to mention that approaches like the latter *are* available solutions and frame the contribution of the paper rather as one of providing a "better" solution in whichever way the authors feel this is best described (more-efficient, etc.). MINOR COMMENTS: * It seems that else at the beginning of Algorithm 3, line 9 doesn't belong there. * Several times in the paper, it is mentioned that experiments are done in "two environments," but aren't there three? * The definition in (2) is odd given that you say the "budget evolves as part of the dynamics." That is, if \beta'=\beta_a (as suggested by the Dirac function, then it is merely whatever the action says it was, correct? Why is that part of "the dynamics?"

Reviewer 3



Although building heavily on previous work (Boutilier and Lu, 2016), this paper makes a novel effort in proposing the BMDP framework as a concrete solution to safe RL. This includes theoretical results (both positive and negative) and practical algorithms. The paper is well written, but could be organized better (e.g., page 3 is just a sequence of formal statements without any cohesive text). Clarity could be improved: since the notation is particularly heavy, important quantities should be called by their name more often. My main issue is with the presentation of the results on the optimality operator. The fact that T is, in general, not a contraction, should be discussed in more depth in the paper, since it determines the legitimacy of the later proposed algorithms. In fact, it is shown in the appendix that T is a contraction when restricted to "smooth" value functions. Unfortunately, this is presented in the paper as a mere conjecture, for reasons I do not understand. Other issues: 1. The exploration strategy from Section 3.2 is merely described, without enough intuition on why it is needed and how it is expected to solve the problem 2. The implementation proposed in Section 4 seems to rely heavily on results from Boutilier and Lu, 2016, but a lot of the necessary context is missing. Moreover, a high-level description of Algorithm 3 would be useful. --- After discussion: given the new theorem, I raised the score. I expect the authors to improve the presentation of the proposed algorithm.

[Author Response · NeurIPS 2019]

First of all, we would like to thank the reviewers for their valuable comments, which will help us improve the revised
manuscript. We endeavour to address all the major remarks below.

Second, we would like to notify all the reviewers about a major theoretical improvement allowing us to transform the
conjecture at line 107 into a theorem. At submission time, the proof was missing two steps (lines 443–444 and 465),
and we decided to present it as a conjecture for the sake of caution. In the meantime, we have proved them and we
propose to make it a theorem.

Third, a remark common to all reviewers is that Algorithm 3 (convex hull policy) lacks a detailed explanation, which
we acknowledge: it was a questionable decision due to space constraints. We will include a descriptive paragraph.

**Reviewer #1** **In principle it could be framed [...] why this is not a good idea.** Swapping the min and max would
lead to a different set of policies. Indeed, the BMDP solution is generally not a saddle point: max-min < min-max.

**When creating batch samples what happens when the budget is zero? The episodes ends with a negative reward?**
If the agent explores/exploits with zero budget, it will always select the estimated safest action, with minimal estimated
expected cost $Q_c$. If costs are still incurred, the model $Q_\theta$ will be updated accordingly.

**As it is not a contraction in general, in which cases should we not use the proposed approach?** The Remark 1 and
the counter-example in Theorem 2 both indicate that the proposed approach is more likely to diverge when $Q^*$ is steep,
i.e. in problems where a small increase in budget $\beta$ leads to substantial gains in rewards. As in most theoretical works,
the required assumptions may be transgressed in real world. But, it does not necessarily imply that the algorithm would
not work in practice. For a sensitive real world application, we would not refrain from using the algorithm, but would
strongly advise to supervise it.

**How can the budget lie between two $Q_c$, is it not the case that all of them should respect the budget constraint?**
$Q_c^*(\overline{s}, \overline{a})$ is the cost induced by *first* executing action $\overline{a}$ and only *then* following the optimal budgeted policy $\pi^*$. Among
all these actions, some of them may not be feasible and exceed the budget $\beta$, which is why an additional optimisation
step is required in the definition of the greedy policy.

**Reviewer #2** **(A.1)** The cost constraints only apply to the final trained budgeted policy, we indeed do not care about
the costs incurred during training. To enforce a specific $\beta$ at test time, set the initial augmented state to $\overline{s}_0 = (s_0, \beta)$.

**(A.2)** If we reach Line 9 in Algorithm 3, it must be that the condition $\beta < q_c^2$ in Line 5 never holds, which means that
every action verifies $q_c \leq \beta$: it is a case where the budget is always respected.

**(A.3)** We will follow your suggestion: indeed Algorithm 4 is more specific to our work than Algorithms 1/2.

**(B)** We will make the introduction clearer and properly introduce the "set of solutions" techniques like FTQ($\lambda$) along
with a proper comparison to the BMDP approach. Namely, they do not recover optimal budgeted policies, in addition to
being inefficient.

**Minor comment 2** Indeed, but one of them is just a toy example to illustrate the risk-sensitive exploration only.

**Minor comment 3** This is absolutely correct. The problem is that since we framed the problem in an augmented space
where the budget $\beta$ is part of the state $\overline{s}$, its evolution can only be described within the dynamics of $\overline{s}$ (which we specify).
We agree that this makes the notations quite obscure for a very simple idea, but on the other hand casting the problem in
such a way drastically simplifies the next definitions and proofs, e.g. that of Proposition 1.

**Reviewer #3** **Presentation of the result on the optimality operator.** The main criticism concerned the presenta-
tion of the contractivity of $\mathcal{T}$ over smooth Q-functions as a conjecture rather than a proven result. We hope that these
reservations have been lifted by the upgrade of the conjecture to a theorem.

**The exploration strategy from Section 3.2 is merely described**: This strategy is motivated by the observation that
a wide variety of risk levels needs to be experienced during training, which can be achieved by enforcing the risk
constraints during data collection. This intuition was meant to be conveyed by the corridor example where a conventional
greedy exploration procedure fails to visit the safe region. We will add an additional discussion to clarify this point.

**The implementation proposed in Section 4 seems to rely heavily on results from Boutilier and Lu, 2016, but a
lot of the necessary context is missing**. Several differences exist between the approach from Boutilier and Lu (2016) -
the greedy budget allocation (GBA) algorithm - and ours: they rely on a finite set of non-dominated budget points $B$
which grows exponentially with the horizon and becomes uncountably infinite in continuous state spaces. They also
compute and sort a matrix of bang-per-buck ratios of size $|S| \times B$, which is again infeasible when $S$ is continuous. In
contrast, we instead rely on estimating the optimal cost-to-go $Q_c^*$, which requires an additional min constraint in its
definition that does not appear in Boutilier and Lu (2016) (they only estimate $V_r^*$).

[Meta-Review · NeurIPS 2019]

The paper formulates a budgeted Markov decision process (BMDP) able to deal with large search spaces. All reviewers feel the proposed method is novel, interesting and could be an important step in trying to address some existing problems with "modern" RL approaches.